# Stochastic Variance Reduction Methods
# for Saddle-Point Problems

**P. Balamurugan**
INRIA - Ecole Normale Supérieure, Paris
`balamurugan.palaniappan@inria.fr`

**Francis Bach**
INRIA - Ecole Normale Supérieure, Paris
`francis.bach@ens.fr`

## Abstract

We consider convex-concave saddle-point problems where the objective functions may be split in many components, and extend recent stochastic variance reduction methods (such as SVRG or SAGA) to provide the first large-scale linearly convergent algorithms for this class of problems which are common in machine learning. While the algorithmic extension is straightforward, it comes with challenges and opportunities: (a) the convex minimization analysis does not apply and we use the notion of monotone operators to prove convergence, showing in particular that the same algorithm applies to a larger class of problems, such as variational inequalities, (b) there are two notions of splits, in terms of functions, or in terms of partial derivatives, (c) the split does need to be done with convex-concave terms, (d) non-uniform sampling is key to an efficient algorithm, both in theory and practice, and (e) these incremental algorithms can be easily accelerated using a simple extension of the "catalyst" framework, leading to an algorithm which is always superior to accelerated batch algorithms.

## 1 Introduction

When using optimization in machine learning, leveraging the natural separability of the objective functions has led to many algorithmic advances; the most common example is the separability as a sum of individual loss terms corresponding to individual observations, which leads to stochastic gradient descent techniques. Several lines of work have shown that the plain Robbins-Monro algorithm could be accelerated for strongly-convex finite sums, e.g., SAG [1], SVRG [2], SAGA [3]. However, these only apply to separable objective functions.

In order to tackle non-separable losses or regularizers, we consider the saddle-point problem:

$$\min_{x \in \mathbb{R}^d} \max_{y \in \mathbb{R}^n} \quad K(x, y) + M(x, y), \tag{1}$$

where the functions $K$ and $M$ are "convex-concave", that is, convex with respect to the first variable, and concave with respect to the second variable, with $M$ potentially non-smooth but "simple" (e.g., for which the proximal operator is easy to compute), and $K$ smooth. These problems occur naturally within convex optimization through Lagrange or Fenchel duality [4]; for example the bilinear saddle-point problem $\min_{x \in \mathbb{R}^d} \max_{y \in \mathbb{R}^n} f(x) + y^\top K x - g(y)$ corresponds to a supervised learning problem with design matrix $K$, a loss function $g^*$ (the Fenchel conjugate of $g$) and a regularizer $f$.

We assume that the function $K$ may be split into a potentially large number of components. Many problems in machine learning exhibit that structure in the saddle-point formulation, but not in the associated convex minimization and concave maximization problems (see examples in Section 2.1).

Like for convex minimization, gradient-based techniques that are blind to this separable structure need to access all the components at every iteration. We show that algorithms such as SVRG [2] and SAGA [3] may be readily extended to the saddle-point problem. While the algorithmic extension is straightforward, it comes with challenges and opportunities. We make the following contributions:

- We provide the first convergence analysis for these algorithms for saddle-point problems, which differs significantly from the associated convex minimization set-up. In particular, we use in Section 6 the interpretation of saddle-point problems as finding the zeros of a monotone operator, and only use the monotonicity properties to show linear convergence of our algorithms, thus showing that they extend beyond saddle-point problems, e.g., to variational inequalities [5, 6].

- We show that the saddle-point formulation (a) allows two different notions of splits, in terms of functions, or in terms of partial derivatives, (b) does need splits into convex-concave terms (as opposed to convex minimization), and (c) that non-uniform sampling is key to an efficient algorithm, both in theory and practice (see experiments in Section 7).

- We show in Section 5 that these incremental algorithms can be easily accelerated using a simple extension of the "catalyst" framework of [7], thus leading to an algorithm which is always superior to accelerated batch algorithms.

## 2  Composite Decomposable Saddle-Point Problems

We now present our new algorithms on saddle-point problems and show a natural extension to monotone operators later in Section 6. We thus consider the saddle-point problem defined in Eq. (1), with the following assumptions:

(A) $M$ is strongly $(\lambda, \gamma)$-convex-concave, that is, the function $(x, y) \mapsto M(x, y) - \frac{\lambda}{2}\|x\|^2 + \frac{\gamma}{2}\|y\|^2$ is convex-concave. Moreover, we assume that we may compute the proximal operator of $M$, i.e., for any $(x', y') \in \mathbb{R}^{n+d}$ ($\sigma$ is the step-length parameter associated with the prox operator):

$$\mathrm{prox}_M^\sigma(x', y') = \arg\min_{x \in \mathbb{R}^d} \max_{y \in \mathbb{R}^n} \ \sigma M(x, y) + \frac{1}{2}\|x - x'\|^2 - \frac{\gamma}{2}\|y - y'\|^2. \qquad (2)$$

The values of $\lambda$ and $\gamma$ lead to the definition of a weighted Euclidean norm on $\mathbb{R}^{n+d}$ defined as $\Omega(x, y)^2 = \lambda\|x\|^2 + \gamma\|y\|^2$, with dual norm defined through $\Omega^*(x, y)^2 = \lambda^{-1}\|x\|^2 + \gamma^{-1}\|y\|^2$. Dealing with the two different scaling factors $\lambda$ and $\gamma$ is crucial for good performance, as these may be very different, depending on the many arbitrary ways to set-up a saddle-point problem.

(B) $K$ is convex-concave and has Lipschitz-continuous gradients; it is natural to consider the *gradient operator* $B : \mathbb{R}^{n+d} \to \mathbb{R}^{n+d}$ defined as $B(x, y) = (\partial_x K(x, y), -\partial_y K(x, y)) \in \mathbb{R}^{n+d}$ and to consider $L = \sup_{\Omega(x-x', y-y')=1} \Omega^*(B(x, y) - B(x', y'))$. The quantity $L$ represents the condition number of the problem.

(C) The vector-valued function $B(x, y) = (\partial_x K(x, y), -\partial_y K(x, y)) \in \mathbb{R}^{n+d}$ may be split into a family of vector-valued functions as $B = \sum_{i \in \mathcal{I}} B_i$, where the only constraint is that each $B_i$ is Lipschitz-continuous (with constant $L_i$). There is no need to assume the existence of a function $K_i : \mathbb{R}^{n+d} \to \mathbb{R}$ such that $B_i = (\partial_x K_i, -\partial_y K_i)$.

   We will also consider splits which are adapted to the saddle-point nature of the problem, that is, of the form $B(x, y) = \big(\sum_{k \in \mathcal{K}} B_k^x(x, y), \sum_{j \in \mathcal{J}} B_j^y(x, y)\big)$, which is a subcase of the above with $\mathcal{I} = \mathcal{J} \times \mathcal{K}$, $B_{jk}(x, y) = (p_j B_k^x(x, y), q_k B_j^y(x, y))$, for $p$ and $q$ sequences that sum to one. This substructure, which we refer to as "factored", will only make a difference when storing the values of these operators in Section 4 for our SAGA algorithm.

Given assumptions (A)-(B), the saddle-point problem in Eq. (1) has a unique solution $(x_*, y_*)$ such that $K(x_*, y) + M(x_*, y) \leqslant K(x_*, y_*) + M(x_*, y_*) \leqslant K(x, y_*) + M(x, y_*)$, for all $(x, y)$; moreover $\min_{x \in \mathbb{R}^d} \max_{y \in \mathbb{R}^n} K(x, y) + M(x, y) = \max_{y \in \mathbb{R}^n} \min_{x \in \mathbb{R}^d} K(x, y) + M(x, y)$ (see, e.g., [8, 4]).

The main generic examples for the functions $K(x, y)$ and $M(x, y)$ are:

- **Bilinear saddle-point problems**: $K(x, y) = y^\top K x$ for a matrix $K \in \mathbb{R}^{n \times d}$ (we identify here a matrix with the associated bilinear function), for which the vector-valued function $B(x, y)$ is linear, i.e., $B(x, y) = (K^\top y, -Kx)$. Then $L = \|K\|_{\mathrm{op}}/\sqrt{\gamma\lambda}$, where $\|K\|_{\mathrm{op}}$ is the largest singular value of $K$.

   There are two natural potential splits with $\mathcal{I} = \{1, \ldots, n\} \times \{1, \ldots, d\}$, with $B = \sum_{j=1}^n \sum_{k=1}^d B_{jk}$: (a) the split into individual elements $B_{jk}(x, y) = K_{jk}(y_j, -x_k)$, where every element is the gradient operator of a bi-linear function, and (b) the "factored" split into rows/columns $B_{jk}(x, y) = (q_k y_j K_{j\cdot}^\top, -p_j x_k K_{\cdot k})$, where $K_{j\cdot}$ and $K_{\cdot k}$ are the $j$-th row and $k$-th column of $K$, $p$ and $q$ are any set of vectors summing to one, and every element is not the gradient operator of any function. These splits correspond to several "sketches" of the matrix $K$ [9], adapted to subsampling of $K$, but other sketches could be considered.

– **Separable functions**: $M(x, y) = f(x) - g(y)$ where $f$ is any $\lambda$-strongly-convex and $g$ is $\gamma$-strongly convex, for which the proximal operators $\text{prox}_f^\sigma(x') = \arg\min_{x \in \mathbb{R}^d} \sigma f(x) + \frac{\lambda}{2}\|x - x'\|^2$ and $\text{prox}_g^\sigma(y') = \arg\max_{y \in \mathbb{R}^d} -\sigma g(y) - \frac{\gamma}{2}\|y - y'\|^2$ are easy to compute. In this situation, $\text{prox}_M^\sigma(x', y') = (\text{prox}_f^\sigma(x'), \text{prox}_g^\sigma(y'))$. Following the usual set-up of composite optimization [10], no smoothness assumption is made on $M$ and hence on $f$ or $g$.

## 2.1 Examples in machine learning

Many learning problems are formulated as convex optimization problems, and hence by duality as saddle-point problems. We now give examples where our new algorithms are particularly adapted.

**Supervised learning with non-separable losses or regularizers.** For regularized linear supervised learning, with $n$ $d$-dimensional observations put in a design matrix $K \in \mathbb{R}^{n \times d}$, the predictions are parameterized by a vector $x \in \mathbb{R}^d$ and lead to a vector of predictions $Kx \in \mathbb{R}^n$. Given a loss function defined through its Fenchel conjugate $g^*$ from $\mathbb{R}^n$ to $\mathbb{R}$, and a regularizer $f(x)$, we obtain exactly a bi-linear saddle-point problem. When the loss $g^*$ or the regularizer $f$ is separable, i.e., a sum of functions of individual variables, we may apply existing fast gradient-techniques [1, 2, 3] to the primal problem $\min_{x \in \mathbb{R}^d} g^*(Kx) + f(x)$ or the dual problem $\max_{y \in \mathbb{R}^n} -g(y) - f^*(K^\top y)$, as well as methods dedicated to separable saddle-point problems [11, 12]. When the loss $g^*$ and the regularizer $f$ are not separable (but have a simple proximal operator), our new fast algorithms are the only ones that can be applied from the class of large-scale linearly convergent algorithms.

Non-separable *losses* may occur when (a) predicting by affine functions of the inputs and not penalizing the constant terms (in this case defining the loss functions as the minimum over the constant term, which becomes non-separable) or (b) using structured output prediction methods that lead to convex surrogates to the area under the ROC curve (AUC) or other precision/recall quantities [13, 14]. These come often with efficient proximal operators (see Section 7 for an example).

Non-separable *regularizers* with available efficient proximal operators are numerous, such as grouped-norms with potentially overlapping groups, norms based on submodular functions, or total variation (see [15] and references therein, and an example in Section 7).

**Robust optimization.** The framework of robust optimization [16] aims at optimizing an objective function with uncertain data. Given that the aim is then to minimize the maximal value of the objective function given the uncertainty, this leads naturally to saddle-point problems.

**Convex relaxation of unsupervised learning.** Unsupervised learning leads to convex relaxations which often exhibit structures naturally amenable to saddle-point problems, e.g, for discriminative clustering [17] or matrix factorization [18].

## 2.2 Existing batch algorithms

In this section, we review existing algorithms aimed at solving the composite saddle-point problem in Eq. (1), without using the sum-structure. Note that it is often possible to apply batch algorithms for the associated primal or dual problems (which are not separable in general).

**Forward-backward (FB) algorithm.** The main iteration is

$$
\begin{aligned}
(x_t, y_t) &= \text{prox}_M^\sigma \left[ (x_{t-1}, y_{t-1}) - \sigma \left( \begin{smallmatrix} 1/\lambda & 0 \\ 0 & 1/\gamma \end{smallmatrix} \right) B(x_{t-1}, y_{t-1}) \right] \\
&= \text{prox}_M^\sigma \left( x_{t-1} - \sigma \lambda^{-1} \partial_x K(x_{t-1}, y_{t-1}) + \sigma \gamma^{-1} \partial_y K(x_{t-1}, y_{t-1}) \right).
\end{aligned}
$$

The algorithm aims at simultaneously minimizing with respect to $x$ while maximizing with respect to $y$ (when $M(x, y)$ is the sum of isotropic quadratic terms and indicator functions, we get simultaneous projected gradient descents). This algorithm is known not to converge in general [8], but is linearly convergent for *strongly*-convex-concave problems, when $\sigma = 1/L^2$, with the rate $(1 - 1/(1 + L^2))^t$ [19] (see simple proof in Appendix B.1). This is the one we are going to adapt to stochastic variance reduction.

When $M(x, y) = f(x) - g(y)$, we obtain the two parallel updates $x_t = \text{prox}_f^\sigma \left( x_{t-1} - \lambda^{-1} \sigma \partial_x K(x_{t-1}, y_{t-1}) \right)$ and $y_t = \text{prox}_g^\sigma \left( y_{t-1} + \gamma^{-1} \sigma \partial_y K(x_{t-1}, y_{t-1}) \right)$, which can de done serially by replacing the second one by $y_t = \text{prox}_g^\sigma \left( y_{t-1} + \gamma^{-1} \sigma \partial_y K(x_t, y_{t-1}) \right)$. This is often referred to as the Arrow-Hurwicz method (see [20] and references therein).

**Accelerated forward-backward algorithm.** The forward-backward algorithm may be accelerated by a simple extrapolation step, similar to Nesterov's acceleration for convex minimization [21]. The algorithm from [20], which *only applies to bilinear functions $K$*, and which we extend from separable $M$ to our more general set-up in Appendix B.2, has the following iteration:

$$(x_t, y_t) = \text{prox}_M^\sigma \left[ (x_{t-1}, y_{t-1}) - \sigma \begin{pmatrix} 1/\lambda & 0 \\ 0 & 1/\gamma \end{pmatrix} B(x_{t-1} + \theta(x_{t-1} - x_{t-2}), y_{t-1} + \theta(y_{t-1} - y_{t-2})) \right].$$

With $\sigma = 1/(2L)$ and $\theta = L/(L+1)$, we get an improved convergence rate, where $(1 - 1/(1 + L^2))^t$ is replaced by $(1 - 1/(1 + 2L))^t$. This is always a strong improvement when $L$ is large (ill-conditioned problems), as illustrated in Section 7. Note that our acceleration technique in Section 5 may be extended to get a similar rate for the batch set-up for non-linear $K$.

## 2.3 Existing stochastic algorithms

Forward-backward algorithms have been studied with added noise [22], leading to a convergence rate in $O(1/t)$ after $t$ iterations for strongly-convex-concave problems. In our setting, we replace $B(x, y)$ in our algorithm with $\frac{1}{\pi_i} B_i(x, y)$, where $i \in \mathcal{I}$ is sampled from the probability vector $(\pi_i)_i$ (good probability vectors will depend on the application, see below for bilinear problems). We have $\mathbb{E} B_i(x, y) = B(x, y)$; the main iteration is then

$$(x_t, y_t) = \text{prox}_M^{\sigma_t} \left[ (x_{t-1}, y_{t-1}) - \sigma_t \begin{pmatrix} 1/\lambda & 0 \\ 0 & 1/\gamma \end{pmatrix} \frac{1}{\pi_{i_t}} B_{i_t}(x_{t-1}, y_{t-1}) \right],$$

with $i_t$ selected independently at random in $\mathcal{I}$ with probability vector $\pi$. In Appendix C, we show that using $\sigma_t = 2/(t + 1 + 8\bar{L}(\pi)^2)$ leads to a convergence rate in $O(1/t)$, where $\bar{L}(\pi)$ is a smoothness constant explicited below. For saddle-point problems, it leads to the complexities shown in Table 1. Like for convex minimization, it is fast early on but the performance levels off. Such schemes are typically used in sublinear algorithms [23].

## 2.4 Sampling probabilities, convergence rates and running-time complexities

In order to characterize running-times, we denote by $T(A)$ the complexity of computing $A(x, y)$ for any operator $A$ and $(x, y) \in \mathbb{R}^{n+d}$, while we denote by $T_{\text{prox}}(M)$ the complexity of computing $\text{prox}_M^\sigma(x, y)$. In our motivating example of bilinear functions $K(x, y)$, we assume that $T_{\text{prox}}(M)$ takes times proportional to $n + d$ and getting a single element of $K$ is $O(1)$.

In order to characterize the convergence rate, we need the Lipschitz-constant $L$ (which happens to be the condition number with our normalization) defined earlier as well as a smoothness constant adapted to our sampling schemes:

$$\bar{L}(\pi)^2 = \sup_{(x,y,x',y')} \sum_{i \in \mathcal{I}} \frac{1}{\pi_i} \Omega^*(B_i(x,y) - B_i(x',y'))^2 \text{ such that } \Omega(x - x', y - y')^2 \leqslant 1.$$

We always have the bounds $L^2 \leqslant \bar{L}(\pi)^2 \leqslant \max_{i \in \mathcal{I}} L_i^2 \times \sum_{i \in \mathcal{I}} \frac{1}{\pi_i}$. However, in structured situations (like in bilinear saddle-point problems), we get much improved bounds, as described below.

**Bi-linear saddle-point.** The constant $L$ is equal to $\|K\|_{\text{op}}/\sqrt{\lambda\gamma}$, and we will consider as well the Frobenius norm $\|K\|_F$ defined through $\|K\|_F^2 = \sum_{j,k} K_{jk}^2$, and the norm $\|K\|_{\max}$ defined as $\|K\|_{\max} = \max\{\sup_j (KK^\top)_{jj}^{1/2}, \sup_k (K^\top K)_{kk}^{1/2}\}$. Among the norms above, we always have:

$$\|K\|_{\max} \leqslant \|K\|_{\text{op}} \leqslant \|K\|_F \leqslant \sqrt{\max\{n,d\}}\|K\|_{\max} \leqslant \sqrt{\max\{n,d\}}\|K\|_{\text{op}}, \qquad (3)$$

which allows to show below that some algorithms have better bounds than others.

There are several schemes to choose the probabilities $\pi_{jk}$ (individual splits) and $\pi_{jk} = p_j q_k$ (factored splits). For the factored formulation where we select random rows and columns, we consider the non-uniform schemes $p_j = (KK^\top)_{jj}/\|K\|_F^2$ and $q_k = (K^\top K)_{kk}/\|K\|_F^2$, leading to $\bar{L}(\pi) \leqslant \|K\|_F/\sqrt{\lambda\gamma}$, or uniform, leading to $\bar{L}(\pi) \leqslant \sqrt{\max\{n,d\}}\|K\|_{\max}/\sqrt{\lambda\gamma}$. For the individual formulation where we select random elements, we consider $\pi_{jk} = K_{jk}^2/\|K\|_F^2$, leading to $\bar{L}(\pi) \leqslant \sqrt{\max\{n,d\}}\|K\|_F/\sqrt{\lambda\gamma}$, or uniform, leading to $\bar{L}(\pi) \leqslant \sqrt{nd}\|K\|_{\max}/\sqrt{\lambda\gamma}$ (in these situations, it is important to select several elements simultaneously, which our analysis supports).

We characterize convergence with the quantity $\varepsilon = \Omega(x - x_*, y - y_*)^2/\Omega(x_0 - x_*, y_0 - y_*)^2$, where $(x_0, y_0)$ is the initialization of our algorithms (typically $(0, 0)$ for bilinear saddle-points). In Table 1 we give a summary of the complexity of all algorithms discussed in this paper: we recover the same type of speed-ups as for convex minimization. A few points are worth mentioning:

| Algorithms | Complexity |
|---|---|
| Batch FB | $\log(1/\varepsilon) \times ($ $nd + nd\|K\|_{\mathrm{op}}^2/(\lambda\gamma)$ $)$ |
| Batch FB-accelerated | $\log(1/\varepsilon) \times ($ $nd + nd\|K\|_{\mathrm{op}}/\sqrt{\lambda\gamma}$ $)$ |
| Stochastic FB-non-uniform | $(1/\varepsilon) \times ($ $\max\{n,d\}\|K\|_F^2/(\lambda\gamma)$ $)$ |
| Stochastic FB-uniform | $(1/\varepsilon) \times ($ $nd\|K\|_{\mathrm{max}}^2/(\lambda\gamma)$ $)$ |
| SAGA/SVRG-uniform | $\log(1/\varepsilon) \times ($ $nd + nd\|K\|_{\mathrm{max}}^2/(\lambda\gamma)$ $)$ |
| SAGA/SVRG-non-uniform | $\log(1/\varepsilon) \times ($ $nd + \max\{n,d\}\|K\|_F^2/(\lambda\gamma)$ $)$ |
| SVRG-non-uniform-accelerated | $\log(1/\varepsilon) \times ($ $nd + \sqrt{nd\max\{n,d\}}\|K\|_F/\sqrt{\lambda\gamma}$ $)$ |

Table 1: Summary of convergence results for the strongly $(\lambda,\gamma)$-convex-concave bilinear saddle-point problem with matrix $K$ and individual splits (and $n+d$ updates per iteration). For factored splits (little difference), see Appendix D.4. For accelerated SVRG, we omitted the logarithmic term (see Section 5).

– Given the bounds between the various norms on $K$ in Eq. (3), SAGA/SVRG with non-uniform sampling always has convergence bounds superior to SAGA/SVRG with uniform sampling, which is always superior to batch forward-backward. Note however, that in practice, SAGA/SVRG with uniform sampling may be inferior to accelerated batch method (see Section 7).

– Accelerated SVRG with non-uniform sampling is the most efficient method, which is confirmed in our experiments. Note that if $n = d$, our bound is better than or equal to accelerated forward-backward, in exactly the same way than for regular convex minimization. There is thus a formal advantage for variance reduction.

## 3   SVRG: Stochastic Variance Reduction for Saddle Points

Following [2, 24], we consider a stochastic-variance reduced estimation of the finite sum $B(x,y) = \sum_{i\in\mathcal{I}} B_i(x,y)$. This is achieved by assuming that we have an iterate $(\tilde{x}, \tilde{y})$ with a known value of $B(\tilde{x}, \tilde{y})$, and we consider the estimate of $B(x,y)$:

$$B(\tilde{x}, \tilde{y}) + \tfrac{1}{\pi_i} B_i(x,y) - \tfrac{1}{\pi_i} B_i(\tilde{x}, \tilde{y}),$$

which has the correct expectation when $i$ is sampled from $\mathcal{I}$ with probability $\pi$, but with a reduced variance. Since we need to refresh $(\tilde{x}, \tilde{y})$ regularly, the algorithm works in epochs (we allow to sample $m$ elements per updates, i.e., a mini-batch of size $m$), with an algorithm that shares the same structure than SVRG for convex minimization; see Algorithm 1. Note that we provide an explicit number of iterations per epoch, proportional to $(L^2 + 3\bar{L}^2/m)$. We have the following theorem, shown in Appendix D.1 (see also a discussion of the proof in Section 6).

**Theorem 1** *Assume (A)-(B)-(C). After $v$ epochs of Algorithm 1, we have:*

$$\mathbb{E}\big[\Omega(x_v - x_*, y_v - y_*)^2\big] \leqslant (3/4)^v \Omega(x_0 - x_*, y_0 - y_*)^2.$$

The computational complexity to reach precision $\varepsilon$ is proportional to $\big[T(B) + (mL^2 + \bar{L}^2)\max_{i\in\mathcal{I}} T(B_i) + (1 + L^2 + \bar{L}^2/m)T_{\mathrm{prox}}(M)\big] \log\frac{1}{\varepsilon}$. Note that by taking the mini-batch size $m$ large, we can alleviate the complexity of the proximal operator $\mathrm{prox}_M$ if too large. Moreover, if $L^2$ is too expensive to compute, we may replace it by $\bar{L}^2$ but with a worse complexity bound.

**Bilinear saddle-point problems.** When using a mini-batch size $m = 1$ with the factored updates, or $m = n + d$ for the individual updates, we get the same complexities proportional to $[nd + \max\{n,d\}\|K\|_F^2/(\lambda\gamma)] \log(1/\varepsilon)$ for non-uniform sampling, which improve significantly over (non-accelerated) batch methods (see Table 1).

## 4   SAGA: Online Stochastic Variance Reduction for Saddle Points

Following [3], we consider a stochastic-variance reduced estimation of $B(x,y) = \sum_{i\in\mathcal{I}} B_i(x,y)$. This is achieved by assuming that we store values $g^i = B_i(x^{\mathrm{old}(i)}, y^{\mathrm{old}(i)})$ for an old iterate

**Algorithm 1** SVRG: Stochastic Variance Reduction for Saddle Points

---

**Input:** Functions $(K_i)_i$, $M$, probabilities $(\pi_i)_i$, smoothness $\bar{L}(\pi)$ and $L$, iterate $(x, y)$
number of epochs $v$, number of updates per iteration (mini-batch size) $m$

   Set $\sigma = \left[L^2 + 3\bar{L}^2/m\right]^{-1}$
   **for** $u = 1$ to $v$ **do**
      Initialize $(\tilde{x}, \tilde{y}) = (x, y)$ and compute $B(\tilde{x}, \tilde{y})$
      **for** $k = 1$ to $\log 4 \times (L^2 + 3\bar{L}^2/m)$ **do**
         Sample $i_1, \ldots, i_m \in \mathcal{I}$ from the probability vector $(\pi_i)_i$ with replacement
         $(x, y) \leftarrow \mathrm{prox}_M^\sigma\left[(x, y) - \sigma\left(\begin{smallmatrix} 1/\lambda & 0 \\ 0 & 1/\gamma \end{smallmatrix}\right)\left(B(\tilde{x}, \tilde{y}) + \frac{1}{m}\sum_{k=1}^m \left\{\frac{1}{\pi_{i_k}} B_{i_k}(x, y) - \frac{1}{\pi_{i_k}} B_{i_k}(\tilde{x}, \tilde{y})\right\}\right)\right]$
      **end for**
   **end for**
**Output:** Approximate solution $(x, y)$

---

$(x^{\mathrm{old}(i)}, y^{\mathrm{old}(i)})$, and we consider the estimate of $B(x, y)$:

$$\sum_{j \in \mathcal{I}} g^j + \frac{1}{\pi_i} B_i(x, y) - \frac{1}{\pi_i} g^i,$$

which has the correct expectation when $i$ is sampled from $\mathcal{I}$ with probability $\pi$. At every iteration, we also refresh the operator values $g^i \in \mathbb{R}^{n+d}$, for the same index $i$ or with a new index $i$ sampled uniformly at random. This leads to Algorithm 2, and we have the following theorem showing linear convergence, proved in Appendix D.2. Note that for bi-linear saddle-points, the initialization at $(0, 0)$ has zero cost (which is not possible for convex minimization).

**Theorem 2** *Assume (A)-(B)-(C). After $t$ iterations of Algorithm 2 (with the option of resampling when using non-uniform sampling), we have:*

$$\mathbb{E}\left[\Omega(x_t - x_*, y_t - y_*)^2\right] \leqslant 2\left(1 - (\max\{\tfrac{3|\mathcal{I}|}{2m}, 1 + \tfrac{L^2}{\mu^2} + 3\tfrac{\bar{L}^2}{m\mu^2}\})^{-1}\right)^t \Omega(x_0 - x_*, y_0 - y_*)^2.$$

**Resampling or re-using the same gradients.** For the bound above to be valid for non-uniform sampling, like for convex minimization [25], we need to resample $m$ operators after we make the iterate update. In our experiments, following [25], we considered a mixture of uniform and non-uniform sampling, without the resampling step.

**SAGA vs. SVRG.** The difference between the two algorithms is the same as for convex minimization (see, e.g., [26] and references therein), that is SVRG has no storage, but works in epochs and requires slightly more accesses to the oracles, while SAGA is a pure online method with fewer parameters but requires some storage (for bi-linear saddle-point problems, we only need to store $O(n+d)$ elements for the factored splits, while we need $O(dn)$ for the individual splits). Overall they have the same running-time complexity for individual splits; for factored splits, see Appendix D.4.

**Factored splits.** When using factored splits, we need to store the two parts of the operator values separately and update them independently, leading in Theorem 2 to replacing $|\mathcal{I}|$ by $\max\{|\mathcal{I}|, |\mathcal{K}|\}$.

## 5 Acceleration

Following the "catalyst" framework of [7], we consider a sequence of saddle-point problems with added regularization; namely, given $(\bar{x}, \bar{y})$, we use SVRG to solve approximately

$$\min_{x \in \mathbb{R}^d} \max_{y \in \mathbb{R}^n} K(x, y) + M(x, y) + \tfrac{\lambda\tau}{2}\|x - \bar{x}\|^2 - \tfrac{\gamma\tau}{2}\|y - \bar{y}\|^2, \tag{4}$$

for well-chosen $\tau$ and $(\bar{x}, \bar{y})$. The main iteration of the algorithm differs from the original SVRG by the presence of the iterate $(\bar{x}, \bar{y})$, which is updated regularly (after a precise number of epochs), and different step-sizes (see details in Appendix D.3). The complexity to get an approximate solution of Eq. (4) (forgetting the complexity of the proximal operator and for a single update), up to logarithmic terms, is proportional, to $T(B) + \bar{L}^2(1 + \tau)^{-2} \max_{i \in \mathcal{I}} T(B_i)$.

The key difference with the convex optimization set-up is that the analysis is simpler, without the need for Nesterov acceleration machinery [21] to define a good value of $(\bar{x}, \bar{y})$; indeed, the solution of Eq. (4) is one iteration of the proximal-point algorithm, which is known to converge

**Algorithm 2** SAGA: Online Stochastic Variance Reduction for Saddle Points
***

**Input:** Functions $(K_i)_i$, $M$, probabilities $(\pi_i)_i$, smoothness $\bar{L}(\pi)$ and $L$, iterate $(x, y)$
   number of iterations $t$, number of updates per iteration (mini-batch size) $m$

 Set $\sigma = \big[\max\{\frac{3|\mathcal{I}|}{2m} - 1, L^2 + 3\frac{\bar{L}^2}{m}\}\big]^{-1}$
 Initialize $g^i = B_i(x, y)$ for all $i \in \mathcal{I}$ and $G = \sum_{i \in \mathcal{I}} g^i$
 **for** $u = 1$ to $t$ **do**
    Sample $i_1, \ldots, i_m \in \mathcal{I}$ from the probability vector $(\pi_i)_i$ with replacement
    Compute $h_k = B_{i_k}(x, y)$ for $k \in \{1, \ldots, m\}$
    $(x, y) \leftarrow \text{prox}_M^\sigma \big[(x, y) - \sigma \big(\begin{smallmatrix} 1/\lambda & 0 \\ 0 & 1/\gamma \end{smallmatrix}\big)\big(G + \frac{1}{m}\sum_{k=1}^m \big\{\frac{1}{\pi_{i_k}} h_k - \frac{1}{\pi_{i_k}} g^{i_k}\big\}\big)\big]$
    (optional) Sample $i_1, \ldots, i_m \in \mathcal{I}$ uniformly with replacement
    (optional) Compute $h_k = B_{i_k}(x, y)$ for $k \in \{1, \ldots, m\}$
    Replace $G \leftarrow G - \sum_{k=1}^m \{g^{i_k} - h_k\}$ and $g^{i_k} \leftarrow h_k$ for $k \in \{1, \ldots, m\}$
 **end for**
**Output:** Approximate solution $(x, y)$
***

linearly [27] with rate $(1 + \tau^{-1})^{-1} = (1 - \frac{1}{1+\tau})$. Thus the overall complexity is up to loga-rithmic terms equal to $T(B)(1 + \tau) + \bar{L}^2(1 + \tau)^{-1} \max_{i \in \mathcal{I}} T(B_i)$. The trade-off in $\tau$ is opti-mal for $1 + \tau = \bar{L}\sqrt{\max_{i \in \mathcal{I}} T(B_i)/T(B)}$, showing that there is a potential acceleration when $\bar{L}\sqrt{\max_{i \in \mathcal{I}} T(B_i)/T(B)} \geqslant 1$, leading to a complexity $\bar{L}\sqrt{T(B)\max_{i \in \mathcal{I}} T(B_i)}$.

Since the SVRG algorithm already works in epochs, this leads to a simple modification where every $\log(1 + \tau)$ epochs, we change the values of $(\bar{x}, \bar{y})$. See Algorithm 3 in Appendix D.3. Moreover, we can adaptively update $(\bar{x}, \bar{y})$ more aggressively to speed-up the algorithm.

The following theorem gives the convergence rate of the method (see proof in Appendix D.3). With the value of $\tau$ defined above (corresponding to $\tau = \max\big\{0, \frac{\|K\|_F}{\sqrt{\lambda\gamma}}\sqrt{\max\{n^{-1}, d^{-1}\}} - 1\big\}$ for bilinear problems), we get the complexity $\bar{L}\sqrt{T(B)\max_{i \in \mathcal{I}} T(B_i)}$, up to the logarithmic term $\log(1 + \tau)$. For bilinear problems, this provides a significant acceleration, as shown in Table 1.

**Theorem 3** *Assume (A)-(B)-(C). After $v$ epochs of Algorithm 3, we have, for any positive $v$:*

$$\mathbb{E}\big[\Omega(x_v - x_*, y_v - y_*)^2\big] \leqslant \big(1 - \tfrac{1}{\tau+1}\big)^v \Omega(x_0 - x_*, y_0 - y_*)^2.$$

While we provide a proof only for SVRG, the same scheme should work for SAGA. Moreover, the same idea also applies to the batch setting (by simply considering $|\mathcal{I}| = 1$, i.e., a single function), leading to an acceleration, but now valid for all functions $K$ (not only bilinear).

## 6   Extension to Monotone Operators

In this paper, we have chosen to focus on saddle-point problems because of their ubiquity in machine learning. However, it turns out that our algorithm and, more importantly, our analysis extend to all set-valued *monotone operators* [8, 28]. We thus consider a maximal strongly-monotone operator $A$ on a Euclidean space $\mathcal{E}$, as well as a finite family of Lipschitz-continuous (not necessarily monotone) operators $B_i$, $i \in \mathcal{I}$, with $B = \sum_{i \in \mathcal{I}} B_i$ monotone. Our algorithm then finds the zeros of $A + \sum_{i \in \mathcal{I}} B_i = A + B$, from the knowledge of the resolvent ("backward") operator $(I + \sigma A)^{-1}$ (for a well chosen $\sigma > 0$) and the forward operators $B_i$, $i \in \mathcal{I}$. Note the difference with [29], which requires each $B_i$ to be monotone with a known resolvent and $A$ to be monotone and single-valued. There several interesting examples (on which our algorithms apply):

– **Saddle-point problems**: We assume for simplicity that $\lambda = \gamma = \mu$ (this can be achieved by a simple change of variable). If we denote $B(x, y) = (\partial_x K(x, y), -\partial_y K(x, y))$ and the multi-valued operator $A(x, y) = (\partial_x M(x, y), -\partial_y M(x, y))$, then the proximal operator $\text{prox}_M^\sigma$ may be written as $(\mu I + \sigma A)^{-1}(\mu x, \mu y)$, and we recover exactly our framework from Section 2.

– **Convex minimization**: $A = \partial g$ and $B_i = \partial f_i$ for a strongly-convex function $g$ and smooth func-tions $f_i$: we recover proximal-SVRG [24] and SAGA [3], to minimize $\min_{z \in \mathcal{E}} g(z) + \sum_{i \in \mathcal{I}} f_i(z)$. However, this is a situation where the operators $B_i$ have an extra property called co-coercivity [6],

which we are not using because it is not satisfied for saddle-point problems. The extension of SAGA and SVRG to monotone operators was proposed earlier by [30], but only co-coercive operators are considered, and thus only convex minimization is considered (with important extensions beyond plain SAGA and SVRG), while our analysis covers a much broader set of problems. In particular, the step-sizes obtained with co-coercivity lead to divergence in the general setting.

Because we do not use co-coercivity, applying our results directly to convex minimization, we would get slower rates, while, as shown in Section 2.1, they can be easily cast as a saddle-point problem if the proximal operators of the functions $f_i$ are known, and we then get the same rates than existing fast techniques which are dedicated to this problem [1, 2, 3].

– **Variational inequality problems**, which are notably common in game theory (see, e.g., [5]).

## 7 Experiments

We consider binary classification problems with design matrix $K$ and label vector in $\{-1, 1\}^n$, a non-separable strongly-convex regularizer with an efficient proximal operator (the sum of the squared norm $\lambda \|x\|^2/2$ and the clustering-inducing term $\sum_{i \neq j} |x_i - x_j|$, for which the proximal operator may be computed in $O(n \log n)$ by isotonic regression [31]) and a non-separable smooth loss (a surrogate to the area under the ROC curve, defined as proportional to $\sum_{i_+ \in I_+} \sum_{i_- \in I_-} (1 - y_i + y_j)^2$, where $I_+/I_-$ are sets with positive/negative labels, for a vector of prediction $y$, for which an efficient proximal operator may be computed as well, see Appendix E).

Our upper-bounds depend on the ratio $\|K\|_F^2/(\lambda\gamma)$ where $\lambda$ is the regularization strength and $\gamma \approx n$ in our setting where we minimize an average risk. Setting $\lambda = \lambda_0 = \|K\|_F^2/n^2$ corresponds to a regularization proportional to the average squared radius of the data divided by $1/n$ which is standard in this setting [1]. We also experiment with smaller regularization (i.e., $\lambda/\lambda_0 = 10^{-1}$), to make the problem more ill-conditioned (it turns out that the corresponding testing losses are sometimes slightly better). We consider two datasets, sido ($n = 10142$, $d = 4932$, non-separable losses and regularizers presented above) and rcv1 ($n = 20242$, $d = 47236$, separable losses and regularizer described in Appendix F, so that we can compare with SAGA run in the primal). We report below the squared distance to optimizers which appears in our bounds, as a function of the number of passes on the data (for more details and experiments with primal-dual gaps and testing losses, see Appendix F). Unless otherwise specified, we always use non-uniform sampling.

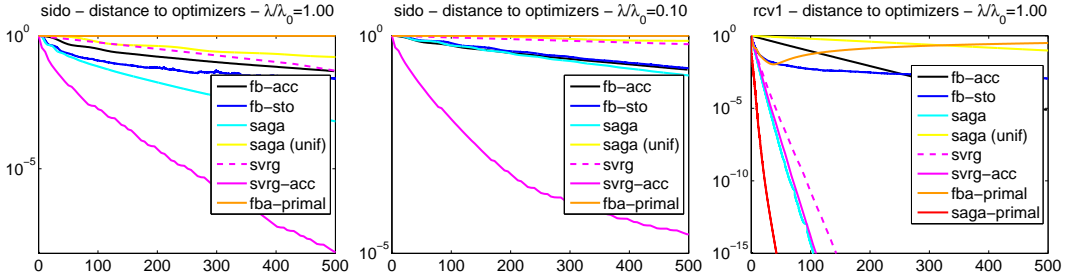

We see that uniform sampling for SAGA does not improve on batch methods, SAGA and accelerated SVRG (with non-uniform sampling) improve significantly over the existing methods, with a stronger gain for the accelerated version for ill-conditioned problems (middle vs. left plot). On the right plot, we compare to primal methods on a separable loss, showing that primal methods (here "fba-primal", which is Nesterov acceleration) that do not use separability (and can thus be applied in all cases) are inferior, while SAGA run on the primal remains faster (but cannot be applied for non-separable losses).

## 8 Conclusion

We proposed the first linearly convergent incremental gradient algorithms for saddle-point problems, which improve both in theory and practice over existing batch or stochastic algorithms. While we currently need to know the strong convexity-concavity constants, we plan to explore in future work adaptivity to these constants like already obtained for convex minimization [3], paving the way to an analysis without strong convexity-concavity.

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
