[Supplementary Material]

# Stochastic Variance Reduction Methods for Saddle-Point Problems
## P. Balamurugan and F. Bach
### *Supplementary material - NIPS 2016*

## A    Formalization through Monotone Operators

Throughout the proofs, we will consider only maximal monotone operators on a Euclidean space $\mathcal{E}$, that is $A$ is assumed to be a $\mu$-strongly monotone (corresponding to $M$ for saddle-points) and potentially set-valued, while $B$ is monotone and $L$-Lipschitz-continuous with respect to the Euclidean norm (and hence single-valued). For an introduction to monotone operators, see [8, 28].

For simplicity, in this appendix, we will only consider a single-valued operator $A$ (noting that the proof extends to any set-valued operator $A$), and we will mostly focus here on the monotonicity properties (noting that the "maximal" property can be treated rigorously [8], in particular to ensure that the resolvent operator is defined everywhere). An operator is monotone if and only if for all $(z, z')$, $(A(z) - A(z'))^\top (z - z') \geqslant 0$. The most basic example is the subdifferential of a convex function. In this paper, we focus on saddle-point problems.

**Application to saddle-point problems.** For the saddle-point problems defined in Section 2 of the main paper, where we have $z = (x, y)$, we need to make a change of variable because of the two potentially different scaling factors $\lambda$ and $\gamma$. We consider the operators

$$B(x, y) = (\lambda^{-1/2}\partial_x K(\lambda^{-1/2}x, \gamma^{-1/2}y), -\gamma^{-1/2}\partial_y K(\lambda^{-1/2}x, \gamma^{-1/2}y))$$
$$A(x, y) = (\lambda^{-1/2}\partial_x M(\lambda^{-1/2}x, \gamma^{-1/2}y), -\gamma^{-1/2}\partial_y M(\lambda^{-1/2}x, \gamma^{-1/2}y)).$$

The solutions of $A(x, y) + B(x, y) = 0$ are exactly the solutions of the problem in Eq. (1), rescaled by $\lambda^{1/2}$ and $\gamma^{1/2}$. Moreover, the operator $A$ is $\mu$-monotone with $\mu = 1$, i.e., for any $z, z'$, we have $(A(z) - A(z'))^\top (z - z') \geqslant \|z - z'\|^2$. Finally, our definition of the smoothness constants for $B$ and $B_i$ in the main paper, exactly leads to a Lipschitz-constant of $L$ with respect to the natural Euclidean norm (a similar result holds for the constant $\bar{L}(\pi)$ defined later). Moreover, convergence results in the Euclidean norm here transfer to convergence results in the norm $\Omega$ defined in the main paper. Note that because of our proofs through operators, it is not easily possible to get bounds on the primal and dual gaps.

**Properties of monotone operators and resolvents.** Given a maximal monotone operator $A$, we may define its *resolvent* operator as $z' = (I + \sigma A)^{-1}(z)$, which is defined as finding the unique $z'$ such that $z' + \sigma A(z') = z$. When $A$ is the operator associated to the saddle-point function $M$ as described above, then the resolvent operator is exactly the proximal operator of $M$ defined in Eq. (2) of the main paper. Note that care has to be taken with the scaling factors $\lambda$ and $\gamma$.

We will use the following properties (on top of Lipschitz-continuity) [8, 28]:

– Monotonicity property: for any $(z, z')$, $(B(z) - B(z'), z - z') \geqslant 0$.
– Contractivity of the resolvent operator for $A$ $\mu$-strongly-monotone: for any $(z, z')$, $\|(I + \sigma A)^{-1}(z) - (I + \sigma A)^{-1}(z')\| \leqslant (1 + \sigma\mu)^{-1}\|z - z'\|$.
– Firm non-expansiveness of the resolvent: for any $(z, z')$, $\|(I + \sigma A)^{-1}(z) - (I + \sigma A)^{-1}(z')\|^2 \leqslant (1 + \sigma\mu)^{-1}(z - z')^\top ((I + \sigma A)^{-1}(z) - (I + \sigma A)^{-1}(z'))$.

Moreover, given our strong-monotonicity assumption, $A + B$ has a unique zero $z_* \in \mathcal{E}$.

Finally in order to characterize the running-times, we will consider the complexity $T_{\text{fw}}(B)$ of computing the operator $B$ and the complexity $T_{\text{bw}}(A)$ to compute the resolvent of $A$. For saddle-point problems, these correspond to $T(B)$ and $T_{\text{prox}}(M)$ from the main paper.

## B    Proof for Deterministic Algorithms

All proofs in this section will follow the same principle, by showing that at every step of our algorithms, a certain function (a "Lyapunov" function) is contracted by a factor strictly less than one.

For the forward-backward algorithm, this will be the distance to optimum $\|z_t - z_*\|^2$; while for the accelerated version, it will be different.

## B.1  Forward-backward algorithm

We consider the iteration $z_t = (I + \sigma A)^{-1}(z_{t-1} - \sigma B(z_{t-1}))$, with $B$ being monotone $L$-Lipschitz-continuous and $A$ being $\mu$-strongly monotone. The optimum $z_*$ (i.e., the zero of $A + B$) is invariant by this iteration. Note that this is the analysis of [19] and that we could improve it by putting some of the strong-monotonicity in the operator $B$ rather than in $A$.

We have:

$$
\begin{aligned}
&\|z_t - z_*\|^2 \\
\leqslant\ & \frac{1}{(1 + \sigma\mu)^2}\|z_{t-1} - z_* - \sigma(B(z_{t-1}) - B(z_*))\|^2 \text{ by contractivity of the resolvent,} \\
=\ & \frac{1}{(1 + \sigma\mu)^2}\left[\|z_{t-1} - z_*\|^2 - 2\sigma(z_{t-1} - z_*)^\top(B(z_{t-1}) - B(z_*)) + \sigma^2\|B(z_{t-1}) - B(z_*)\|^2\right] \\
\leqslant\ & \frac{1}{(1 + \sigma\mu)^2}(1 + \sigma^2 L^2)\|z_{t-1} - z_*\|^2 \text{ by monotonicity of and Lipschitz-continuity of } B, \\
\leqslant\ & \left(\frac{1 + \sigma^2 L^2}{(1 + \sigma\mu)^2}\right)^t\|z_0 - z_*\|^2, \text{ by applying the recursion } t \text{ times.}
\end{aligned}
$$

Thus we get linear (i.e., geometric) convergence as soon as $1 + \sigma^2 L^2 < (1 + \sigma\mu)^2$. If we consider $\eta = \frac{\sigma\mu}{1 + \sigma\mu} \in [0, 1)$, and the rate above becomes equal to:

$$
\frac{1 + \sigma^2 L^2}{(1 + \sigma\mu)^2} = (1 - \eta)^2 + \eta^2\frac{L^2}{\mu^2} = 1 - 2\eta + \eta^2(1 + \frac{L^2}{\mu^2}),
$$

thus the algorithm converges if $\eta < \frac{2}{1 + \frac{L^2}{\mu^2}}$, and with $\eta = \frac{1}{1 + \frac{L^2}{\mu^2}}$ which corresponds to $\sigma = \frac{1}{\mu}\frac{\eta}{1-\eta} = \frac{\mu}{L^2}$, we get a linear convergence rate with constant $1 - \eta = \frac{L^2}{\mu^2 + L^2}$.

Thus the complexity to reach the precision $\varepsilon \times \|z_0 - z_*\|^2$ in squared distance to optimum $\|z_t - z_*\|^2$ is equal to $\left(1 + \frac{L^2}{\mu^2}\right)\left[T_{\text{fw}}(B) + T_{\text{bw}}(A)\right]\log\frac{1}{\varepsilon}$.

Note that we obtain a slow convergence when applied to convex minimization, because we are not using any co-coercivity of $B$, which would lead to a rate $(1 - \mu/L)$ [6]. Indeed, co-coercivity means that $\|B(z) - B(z')\|^2 \leqslant L(B(z) - B(z'))^\top(z - z')$, and this allows to replace above the term $1 + \sigma^2 L^2$ by $1$ if $\sigma \leqslant 2/L$, leading to linear convergence rate with constant $(1 + \mu/L)^{-2} \approx 1 - 2\mu/L$.

## B.2  Accelerated forward-backward algorithm

We consider the iteration $z_t = (I + \sigma A)^{-1}(z_{t-1} - \sigma B[z_{t-1} + \theta(z_{t-1} - z_{t-2})])$, with $B$ being monotone $L$-Lipschitz-continuous and *linear*, and $A$ being $\mu$-strongly monotone. Note that this is an extension of the analysis of [20] to take into account the general monotone operator situation. Again $z_*$ is a fixed-point of the iteration.

Using the firm non-expansiveness of the resolvent operator, we get, with $\eta = \frac{\sigma\mu}{1+\sigma\mu}$, and then using the linearity of $B$:

$$
\begin{aligned}
\|z_t - z_*\|^2 &\leqslant \frac{1}{1+\sigma\mu}(z_t - z_*)^\top \Big[z_{t-1} - z_* - \sigma B[z_{t-1} - z_* + \theta(z_{t-1} - z_{t-2})]\Big]\\
&= (z_t - z_*)^\top \Big[(1-\eta)(z_{t-1} - z_*) - \frac{\eta}{\mu}B[z_{t-1} - z_* + \theta(z_{t-1} - z_{t-2})]\Big]\\
&= -\frac{1-\eta}{2}\|z_t - z_{t-1}\|^2 + \frac{1-\eta}{2}\|z_t - z_*\|^2 + \frac{1-\eta}{2}\|z_{t-1} - z_*\|^2\\
&\qquad\qquad -\frac{\eta}{\mu}(z_t - z_*)^\top B[z_{t-1} - z_* + \theta(z_{t-1} - z_{t-2})]\\
&= -\frac{1-\eta}{2}\|z_t - z_{t-1}\|^2 + \frac{1-\eta}{2}\|z_t - z_*\|^2 + \frac{1-\eta}{2}\|z_{t-1} - z_*\|^2\\
&\qquad\qquad -\frac{\eta}{\mu}(z_t - z_*)^\top B(z_{t-1} - z_*) - \theta\frac{\eta}{\mu}(z_t - z_*)^\top B(z_{t-1} - z_{t-2}),
\end{aligned}
$$

by regrouping terms. By using the Lipschitz-continuity of $B$, we get:

$$
\begin{aligned}
&\|z_t - z_*\|^2\\
&\leqslant -\frac{1-\eta}{2}\|z_t - z_{t-1}\|^2 + \frac{1-\eta}{2}\|z_t - z_*\|^2 + \frac{1-\eta}{2}\|z_{t-1} - z_*\|^2 - \frac{\eta}{\mu}(z_t - z_*)^\top B(z_{t-1} - z_t)\\
&\quad -\theta\frac{\eta}{\mu}(z_{t-1} - z_*)^\top B(z_{t-2} - z_{t-1}) + \theta\frac{\eta}{\mu}L\|z_t - z_{t-1}\|\|z_{t-1} - z_{t-2}\|\\
&\leqslant -\frac{1-\eta}{2}\|z_t - z_{t-1}\|^2 + \frac{1-\eta}{2}\|z_t - z_*\|^2 + \frac{1-\eta}{2}\|z_{t-1} - z_*\|^2 - \frac{\eta}{\mu}(z_t - z_*)^\top B(z_{t-1} - z_t)\\
&\quad -\theta\frac{\eta}{\mu}(z_{t-1} - z_*)^\top B(z_{t-2} - z_{t-1}) + \frac{\theta L}{2}\frac{\eta}{\mu}\Big[\alpha^{-1}\|z_t - z_{t-1}\|^2 + \alpha\|z_{t-1} - z_{t-2}\|^2\Big],
\end{aligned}
$$

with a constant $\alpha > 0$ to be determined later. This leads to, with $\theta = \frac{1-\eta}{1+\eta}$, and by regrouping terms:

$$
\begin{aligned}
&\frac{1+\eta}{2}\|z_t - z_*\|^2 + \Big(\frac{1-\eta}{2} - \frac{\theta\eta L}{2\mu}\alpha^{-1}\Big)\|z_t - z_{t-1}\|^2 - \eta(z_t - z_*)^\top B(z_{t-1} - z_t)\\
&\leqslant \frac{1-\eta}{2}\|z_{t-1} - z_*\|^2 + \Big(\frac{\alpha\eta\theta L}{2\mu}\Big)\|z_{t-1} - z_{t-2}\|^2 - \theta\frac{\eta}{\mu}(z_{t-1} - z_*)^\top B(z_{t-2} - z_{t-1})\\
&\leqslant \theta\Big[\frac{1+\eta}{2}\|z_{t-1} - z_*\|^2 + \Big(\frac{\eta\alpha L}{2\mu}\Big)\|z_{t-1} - z_{t-2}\|^2 - \frac{\eta}{\mu}(z_{t-1} - z_*)^\top B(z_{t-2} - z_{t-1})\Big].
\end{aligned}
$$

We get a Lyapunov function $\mathcal{L} : (z, z') \mapsto \frac{1+\eta}{2}\|z - z_*\|^2 + \Big(\frac{1-\eta}{2} - \frac{\theta\eta L}{2\mu}\alpha^{-1}\Big)\|z - z'\|^2 - \eta(z - z_*)^\top B(z' - z)$, such that $\mathcal{L}(z_t, z_{t-1})$ converges to zero geometrically, if $\frac{\alpha\eta L}{\mu} \leqslant 1 - \eta - \frac{\eta\theta L}{\mu}\alpha^{-1}$ and $\begin{pmatrix} 1+\eta & -\eta L/\mu \\ \eta L/\mu & 1 - \eta - \eta\theta L\mu^{-1}\alpha^{-1} \end{pmatrix} \succcurlyeq 0$. By setting $\eta = \frac{1}{1+2L/\mu}$, and thus $\theta = \frac{1-\eta}{1+\eta} = \frac{1}{1+\mu/L}$, $\sigma = \frac{1}{\mu}\frac{\eta}{1-\eta} = \frac{1}{2L}$, and $\alpha = 1$, we get the desired first property and the fact that the matrix above is greater than $\begin{pmatrix} 1/2 & 0 \\ 0 & 0 \end{pmatrix}$, which allows us to get a linear rate of convergence for $\|z_t - z_*\|^2 \leqslant 2\mathcal{L}(z_t, z_{t-1})$.

## C  Proof for Existing Stochastic Algorithms

We follow [22], but with a specific step-size that leads to a simple result, which also applies to non-uniform sampling from a finite pool. We consider the iteration $z_t = (I + \sigma_t A)^{-1}(z_{t-1} - \sigma_t(Bz_{t-1} + C_t z_{t-1}))$, with $B$ being monotone $L$-Lipschitz-continuous and $A$ being $\mu$-strongly monotone, and $C_t$ a random operator (*not necessarily monotone*) such that $\mathbb{E}C_t(z) = 0$ for all $z$. We assume that all random operators $C_t$ are independent, and we denote by $\mathcal{F}_t$ the $\sigma$-field generated by $C_1, \ldots, C_t$, i.e., the information up to time $t$.

We have with $\mathrm{Lip}(C_t)$ the Lipschitz-constant of $C_t$:

$$\|z_t - z_*\|^2 \;\leqslant\; \frac{1}{(1+\sigma_t\mu)^2}\|z_{t-1} - z_* - \sigma_t(B(z_{t-1}) - B(z_*)) - \sigma_t C_t(z_{t-1})\|^2$$

$$\text{by contractivity of the resolvent,}$$

$$= \frac{1}{(1+\sigma_t\mu)^2}\Big[\|z_{t-1} - z_*\|^2 - 2\sigma_t(z_{t-1} - z_*)^\top(B(z_{t-1}) - B(z_*))$$

$$+ \sigma_t^2\|B(z_{t-1}) - B(z_*) + C_t(z_{t-1})\|^2 + 2\sigma_t(C_t(z_{t-1}))^\top(z_{t-1} - z_*)\Big].$$

By taking conditional expectations, we get:

$$\mathbb{E}\big(\|z_t - z_*\|^2|\mathcal{F}_{t-1}\big) \;\leqslant\; \frac{1}{(1+\sigma_t\mu)^2}\big[(1+\sigma_t^2 L^2)\|z_{t-1} - z_*\|^2 + \sigma_t^2\mathbb{E}(\|C_t(z_{t-1})\|^2|\mathcal{F}_{t-1})\big]$$

$$\text{by monotonicity and Lipschitz-continuity of } B,$$

$$\leqslant \frac{1}{(1+\sigma_t\mu)^2}\big[(1+\sigma_t^2 L^2)\|z_{t-1} - z_*\|^2 + 2\sigma_t^2\mathbb{E}(\|C_t(z_*)\|^2|\mathcal{F}_{t-1})$$

$$+ 2\sigma_t^2\|z_{t-1} - z_*\|^2\mathbb{E}(\sup_{\|z-z'\|=1}\|C_t(z) - C_t(z')\|^2|\mathcal{F}_{t-1})\big]$$

$$= \frac{1}{(1+\sigma_t\mu)^2}\big[(1+\sigma_t^2 L^2)\|z_{t-1} - z_*\|^2 + 2\sigma_t^2\mathbb{E}(\|C_t(z_*)\|^2|\mathcal{F}_{t-1})$$

$$+ 2\sigma_t^2\|z_{t-1} - z_*\|^2\mathbb{E}(\mathrm{Lip}(C_t)^2|\mathcal{F}_{t-1})\big]$$

$$= \frac{1}{(1+\sigma_t\mu)^2}\big[(1+\sigma_t^2 L^2 + 2\sigma_t^2\mathbb{E}(\mathrm{Lip}(C_t)^2|\mathcal{F}_{t-1}))\|z_{t-1} - z_*\|^2 + 2\sigma_t^2\mathbb{E}(\|C_t(z_*)\|^2|\mathcal{F}_{t-1})\big].$$

By denoting $\eta_t = \frac{\sigma_t\mu}{1+\sigma_t\mu} \in [0,1)$, we get

$$\mathbb{E}\|z_t - z_*\|^2 \;\leqslant\; \Big(1 - 2\eta_t + \eta_t^2 + 2\eta_t^2\frac{L^2}{\mu^2} + 2\eta_t^2\frac{1}{\mu^2}\mathbb{E}(\mathrm{Lip}(C_t)^2|\mathcal{F}_{t-1})\Big)\|z_{t-1} - z_*\|^2 + 2\frac{\eta_t^2}{\mu^2}\mathbb{E}(\|C_tz_*\|^2|\mathcal{F}_{t-1})\big].$$

By selecting $\eta_t = \frac{2}{(t+1)+4\frac{L^2}{\mu^2}+\frac{4}{\mu^2}\mathbb{E}(\mathrm{Lip}(C_t)^2|\mathcal{F}_{t-1})} = \frac{2}{t+1+A}$, with $A = 4\frac{L^2}{\mu^2} + \frac{4}{\mu^2}\mathbb{E}(\mathrm{Lip}(C_t)^2|\mathcal{F}_{t-1})$,
we get:

$$\mathbb{E}\|z_t - z_*\|^2 \;\leqslant\; (1 - \eta_t)\mathbb{E}\|z_{t-1} - z_*\|^2 + 2\frac{\eta_t^2}{\mu^2}\mathbb{E}(\|C_tz_*\|^2)\big]$$

$$= \frac{t-1+A}{t+1+A}\mathbb{E}\|z_{t-1} - z_*\|^2 + \frac{8}{(t+1+A)^2}\frac{1}{\mu^2}\mathbb{E}(\|C_tz_*\|^2)$$

$$\leqslant \frac{A(1+A)}{(t+1+A)(t+A)}\|z_0 - z_*\|^2 + \frac{8}{\mu^2}\sum_{u=1}^{t}\frac{(u+A)(u+1+A)}{(t+1+A)(t+A)}\frac{1}{(u+1+A)^2}\mathbb{E}(\|C_uz_*\|^2)$$

$$\text{by expanding the recursion } t \text{ times,}$$

$$\leqslant \frac{A(1+A)}{(t+1+A)(t+A)}\|z_0 - z_*\|^2 + \frac{8}{\mu^2}\sum_{u=1}^{t}\frac{1}{(t+1+A)(t+A)}\mathbb{E}(\|C_uz_*\|^2)$$

$$\leqslant \frac{(1+A)^2}{(t+A)^2}\|z_0 - z_*\|^2 + \frac{8}{\mu^2(t+A)}\sup_{u\in\{1,\ldots,t\}}\mathbb{E}(\|C_uz_*\|^2).$$

The overall convergence rate is in $O(1/t)$ and the constant depends on the noise in the operator values at the optimum. Note that initial conditions are forgotten at a rate $O(1/t^2)$.

**Application to sampling from a finite family.** When sampling from $|\mathcal{I}|$ operators $B_i$, $i \in \mathcal{I}$, and selecting $i_t$ with probability vector $\pi$, then we have $\mathbb{E}(\mathrm{Lip}(C_t)^2|\mathcal{F}_{t-1}) \leqslant \bar{L}(\pi)^2 = \bar{L}^2$ defined as $\sup_{\|z-z'\|\leqslant 1}\sqrt{\sum_{i\in\mathcal{I}}\frac{1}{\pi_i}\|B_i(z) - B_i(z')\|^2}$. Thus, we can take the step-size $\frac{2}{t+1+4\frac{L^2+\bar{L}^2}{\mu^2}}$, which leads to $\sigma_t = \frac{2/\mu}{t+1+4\frac{L^2+\bar{L}^2}{\mu^2}}$. Moreover, if $L$ is unknown (or hard to compute), we can take $\bar{L}$ instead.

We may further bound: $\mathbb{E}(\|C_u z_*\|^2) \leqslant 2\mathbb{E}(\|C_u z_0\|^2) + 2\mathbb{E}(\mathrm{Lip}(C_t)^2)\|z_0 - z_*\|^2$, and thus, if we start from an initial point $z_0$ such that $C_u z_0 = 0$, which is always possible for bi-linear problems, we get an overall bound of (taking $L = \bar{L}$ for simplicity)

$$\Big(\frac{(1 + 8\bar{L}^2/\mu^2)^2}{(t + 8\bar{L}^2/\mu^2)^2} + \frac{16\bar{L}^2/\mu^2}{t + 8\bar{L}^2/\mu^2}\Big)\|z_0 - z_*\|^2 \leqslant \frac{1 + 24\bar{L}^2/\mu^2}{t + 8\bar{L}^2/\mu^2}\|z_0 - z_*\|^2.$$

We thus get an overall $O(1/t)$ convergence rate.

## D   Proof for New Stochastic Algorithms

We also consider the monotone operator set-up, since this is the only assumption that we use. We follow the proof of the corresponding convex minimization algorithms, with key differences which we highlight below. In particular, (a) we do not use function values, and (b) we use shorter step-sizes to tackle the lack of co-coercivity.

### D.1   SVRG: Stochastic-Variance reduced saddle-point problems (Theorem 1)

We only analyze a single epoch starting from the reference estimate $\tilde{z}$, and show that the expected squared distance to optimum is shrunk by a factor of $3/4$ if the number of iterations per epoch is well-chosen. The epoch is started with $z_0 = \tilde{z}$.

We denote by $\mathcal{F}_{t-1}$ the information up to time $t - 1$. We consider sampling $i_{t1}, \ldots, i_{tm} \in \mathcal{I}$ with replacement at time $t$. By using the contractivity of the resolvent operator of $A$, and the fact that $z_* = (I + \sigma A)^{-1}(z_* - \sigma B(z_*))$, we get:

$$\|z_t - z_*\|^2 \quad \leqslant \quad \frac{1}{(1 + \sigma\mu)^2} \Big\| z_{t-1} - z_* - \sigma\Big[B(\tilde{z}) - B(z_*) + \frac{1}{m}\sum_{k=1}^{m} \frac{1}{\pi_{i_{tk}}}(B_{i_{tk}}(z_{t-1}) - B_{i_{tk}}(\tilde{z}))\Big]\Big\|^2$$

$$= \quad \frac{1}{(1 + \sigma\mu)^2} \Big\| z_{t-1} - z_*$$

$$-\sigma\Big[B(z_{t-1}) - B(z_*) + \frac{1}{m}\sum_{k=1}^{m} \frac{1}{\pi_{i_{tk}}}(B_{i_{tk}}(z_{t-1}) - B_{i_{tk}}(\tilde{z})) - (B(z_{t-1}) - B(\tilde{z}))\Big]\Big\|^2.$$

Expanding the squared norm, taking conditional expectations with $\mathbb{E}(\frac{1}{\pi_{i_t k}} B_{i_t k}|\mathcal{F}_{t-1}) = B$, and using the independence of $i_{t1}, \ldots, i_{tm}$, we get:

$$\mathbb{E}\big[\|z_t - z_*\|^2|\mathcal{F}_{t-1}\big]$$

$$\leqslant \quad \frac{1}{(1 + \sigma\mu)^2}\big(\|z_{t-1} - z_*\|^2 - 2\sigma(z_{t-1} - z_*)^\top(B(z_{t-1}) - B(z_*)) + \sigma^2\|B(z_{t-1}) - B(z_*)\|^2\big)$$

$$+\frac{1}{m}\mathbb{E}\Big[\frac{1}{(1 + \sigma\mu)^2}\Big\|\frac{1}{\pi_{i_t}}(B_{i_t}(z_{t-1}) - B_{i_t}(\tilde{z})) - (B(z_{t-1}) - B(\tilde{z}))\Big\|^2\Big|\mathcal{F}_{t-1}\Big].$$

Using the monotonicity of $B$ and the Lipschitz-continuity of $B$ (like in Appendix B.1) , we get the bound

$$\frac{1 + \sigma^2 L^2}{(1 + \sigma\mu)^2}\|z_{t-1} - z_*\|^2 + \frac{1}{m}\mathbb{E}\Big[\frac{1}{(1 + \sigma\mu)^2}\Big\|\frac{1}{\pi_{i_t}}(B_{i_t}(z_{t-1}) - B_{i_t}(\tilde{z})) - (B(z_{t-1}) - B(\tilde{z}))\Big\|^2\Big|\mathcal{F}_{t-1}\Big].$$

We denote by $\bar{L}^2$ the quantity $\bar{L}^2 = \sup_{z,z'\in\mathcal{E}} \frac{1}{\|z-z'\|^2}\sum_{i\in\mathcal{I}} \frac{1}{\pi_i}\|B_i(z) - B_i(z')\|^2$. We then have (using the fact that a variance is less than the second-order moment):

$$\mathbb{E}\Big[\big\|\frac{1}{\pi_{i_t}}(B_{i_t}(z_{t-1}) - B_{i_t}(\tilde{z})) - (B(z_{t-1}) - B(\tilde{z}))\big\|^2\big|\mathcal{F}_{t-1}\Big] \leqslant \mathbb{E}\Big[\big\|\frac{1}{\pi_{i_t}}(B_{i_t}(z_{t-1}) - B_{i_t}(\tilde{z}))\big\|^2\big|\mathcal{F}_{t-1}\Big],$$

which is less than $\bar{L}^2\|z_{t-1} - \tilde{z}\|^2$ because we sample $i_t$ from $\pi$. This leads to

$$\mathbb{E}\big[\|z_t - z_*\|^2|\mathcal{F}_{t-1}\big] \quad \leqslant \quad \frac{1 + \sigma^2 L^2}{(1 + \sigma\mu)^2}\|z_{t-1} - z_*\|^2 + \frac{1}{(1 + \sigma\mu)^2}\frac{\bar{L}^2}{m}\|z_{t-1} - \tilde{z}\|^2$$

$$\leqslant \quad \Big(1 - 2\eta + \eta^2 + \eta^2\frac{L^2}{\mu^2} + \frac{(1 + a)\eta^2}{\mu^2}\frac{\bar{L}^2}{m}\Big)\|z_{t-1} - z_*\|^2$$

$$+\frac{(1 + a^{-1})\eta^2}{\mu^2}\frac{\bar{L}^2}{m}\|\tilde{z} - z_*\|^2,$$

with $\eta = \frac{\sigma\mu}{1+\sigma\mu} \in [0,1)$ and $a > 0$ to be determined later. Assuming that $\eta\left(1 + \frac{L^2}{\mu^2} + \frac{(1+a)}{\mu^2}\frac{\bar{L}^2}{m}\right) \leqslant 1$, and taking full expectations, this leads to:

$$\mathbb{E}\|z_t - z_*\|^2 \leqslant (1-\eta)\mathbb{E}\|z_{t-1} - z_*\|^2 + \frac{(1+a^{-1})\eta^2}{\mu^2}\frac{\bar{L}^2}{m}\|\tilde{z} - z_*\|^2,$$

that is we get a shrinking of the expected distance to optimum with additional noise that depends on the distance to optimum of the reference point $\tilde{z}$. The difference with the convex minimization set-up of [24] is that the proof is more direct, and we get a shrinkage directly on the iterates (we have no choice for monotone operators), without the need to do averaging of the iterates. Moreover, we never use any monotonicity of the operators $B_i$, thus allowing any type of splits (as long as the sum $B$ is monotone).

Then, using the fact that $z_0 = \tilde{z}$, and expanding the recursion:

$$\begin{aligned}
\mathbb{E}\|z_t - z_*\|^2 &\leqslant (1-\eta)^t\|z_0 - z_*\|^2 + \Big(\sum_{u=0}^{t-1}(1-\eta)^u\Big)\frac{(1+a^{-1})\eta^2}{\mu^2}\frac{\bar{L}^2}{m}\|\tilde{z} - z_*\|^2 \\
&\leqslant \left((1-\eta)^t + \frac{(1+a^{-1})\eta}{\mu^2}\frac{\bar{L}^2}{m}\right)\|\tilde{z} - z_*\|^2.
\end{aligned}$$

If we take $a = 2$, $\eta = \frac{1}{1+L^2+3\bar{L}^2/(m\mu^2)}$, which corresponds to $\sigma = \frac{1}{\mu}\frac{\eta}{1-\eta} = \frac{\mu}{L^2+\frac{3}{m}\bar{L}^2}$ and $t = \log 4/\eta = \log 4 \times (1 + \frac{L^2}{\mu^2} + 3\frac{\bar{L}^2}{m\mu^2})$, we obtain a bound of $3/4$, that is, after $t$ steps in an epoch, we obtain $\mathbb{E}\|z_t - z_*\|^2 \leqslant \frac{3}{4}\|\tilde{z} - z_*\|^2$, which is the desired result.

In terms of running-time, we therefore need a time proportional to $T(B) + \left(1 + \frac{L^2}{\mu^2} + 3\frac{\bar{L}^2}{m\mu^2}\right)\left(m \max_i T(B_i) + T_{\text{prox}}(A)\right)$, times $\log\frac{1}{\varepsilon}$ to reach precision $\varepsilon$.

Note that if $L^2$ is too expensive to compute (because it is a global constant), we may replace it by $\bar{L}^2$ and get a worse bound (but still a valid algorithm).

## D.2  SAGA: Online stochastic-variance reduced saddle-point problems (Theorem 2)

The proof follows closely the one of SVRG above. Following the same arguments, we get, by contractivity of the resolvent operator:

$$\begin{aligned}
\|z_t - z_*\|^2 &\leqslant \frac{1}{(1+\sigma\mu)^2}\Big\|z_{t-1} - z_* - \sigma\Big[\sum_{i\in\mathcal{I}}g_{t-1}^i - B(z_*) + \frac{1}{m}\sum_{k=1}^m\frac{1}{\pi_{i_{tk}}}(B_{i_{tk}}(z_{t-1}) - g_{t-1}^{i_{tk}})\Big]\Big\|^2 \\
&= \frac{1}{(1+\sigma\mu)^2}\Big\|z_{t-1} - z_* - \sigma\Big[B(z_{t-1}) - B(z_*) \\
&\qquad\qquad + \frac{1}{m}\sum_{k=1}^m\frac{1}{\pi_{i_{tk}}}(B_{i_{tk}}(z_{t-1}) - g_{t-1}^{i_{tk}}) - (B(z_{t-1}) - \sum_{i\in\mathcal{I}}g_{t-1}^i)\Big]\Big\|^2.
\end{aligned}$$

Then, using independence, monotonicity and Lipschitz-continuity of $B$, we get (note that we never use any monotonicity of $B_i$), like in the proof of Theorem 1:

$$\begin{aligned}
\mathbb{E}\big[\|z_t - z_*\|^2|\mathcal{F}_{t-1}\big] &\leqslant \frac{1+\sigma^2 L^2}{(1+\sigma\mu)^2}\|z_{t-1} - z_*\|^2 \\
&\quad + \frac{1}{m}\mathbb{E}\Big[\frac{1}{(1+\sigma\mu)^2}\big\|\frac{1}{\pi_{i_t}}(B_{i_t}(z_{t-1}) - g_{t-1}^{i_t}) - (B(z_{t-1}) - \sum_{i\in\mathcal{I}}g_{t-1}^i)\big\|^2\big|\mathcal{F}_{t-1}\Big] \\
&\leqslant \frac{1+\sigma^2 L^2}{(1+\sigma\mu)^2}\|z_{t-1} - z_*\|^2 + \frac{1}{m}\frac{1}{(1+\sigma\mu)^2}\Big(\sum_{i\in\mathcal{I}}\frac{1}{\pi_i}\|B_i(z_{t-1}) - g_{t-1}^i\|^2\Big) \\
&\leqslant \left(1 - 2\eta + \eta^2 + \eta^2\frac{L^2}{\mu^2} + \frac{(1+a)\eta^2}{\mu^2}\frac{\bar{L}^2}{m}\right)\|z_{t-1} - z_*\|^2 \\
&\quad + \frac{(1+a^{-1})\eta^2}{\mu^2 m}\Big(\sum_{i\in\mathcal{I}}\frac{1}{\pi_i}\|B_i(z_*) - g_{t-1}^i\|^2\Big),
\end{aligned}$$

with $\eta = \frac{\sigma\mu}{1+\sigma\mu}$. Assuming $\eta\left(1 + \frac{L^2}{\mu^2} + \frac{(1+a)}{\mu^2}\frac{\bar{L}^2}{m}\right) \leqslant 1$, we get

$$\mathbb{E}\left[\|z_t - z_*\|^2 | \mathcal{F}_{t-1}\right] \leqslant (1-\eta)\|z_{t-1} - z_*\|^2 + \frac{(1+a^{-1})\eta^2}{\mu^2 m}\left(\sum_{i\in\mathcal{I}}\frac{1}{\pi_i}\|B_i(z_*) - g_{t-1}^i\|^2\right).$$

Like in the SVRG proof above, we get a contraction of the distance to optimum, with now an added noise that depends on the difference between our stored operator values and the operator values at the global optimum. We thus need to control this distance by adding the proper factors to a Lyapunov function. Note that we never use any monotonicity of the operators $B_i$, thus allowing any type of splits (as long as the sum $B$ is monotone).

We assume that we update (at most $m$ because we are sampling with replacement and we may sample the same gradient twice) "gradients" $g_t^i$ uniformly at random (when we consider uniform sampling, we can reuse the same gradients as dependence does not impact the bound), by replacing them by $g_t^i = B_i(z_{t-1})$. Thus:

$$\mathbb{E}\left(\sum_{i\in\mathcal{I}}\frac{1}{\pi_i}\|B_i(z_*) - g_t^i\|^2 \Big| \mathcal{F}_{t-1}\right)$$

$$= \mathbb{E}\left(\sum_{i \text{ selected}}\frac{1}{\pi_i}\|B_i(z_*) - B_i(z_{t-1})\|^2 + \sum_{i \text{ non selected}}\frac{1}{\pi_i}\|B_i(z_*) - g_{t-1}^i\|^2 \Big| \mathcal{F}_{t-1}\right)$$

$$= \mathbb{E}\left(\sum_{i \text{ selected}}\frac{1}{\pi_i}\left(\|B_i(z_*) - B_i(z_{t-1})\|^2 - \|B_i(z_*) - g_{t-1}^i\|^2\right) + \sum_{i\in\mathcal{I}}\frac{1}{\pi_i}\|B_i(z_*) - g_{t-1}^i\|^2 \Big| \mathcal{F}_{t-1}\right).$$

Since we sample uniformly *with* replacement, the marginal probabilities of selecting an element $i$ is equal to $\rho = 1 - (1 - \frac{1}{|\mathcal{I}|})^m$. We thus get

$$\mathbb{E}\left(\sum_{i\in\mathcal{I}}\frac{1}{\pi_i}\|B_i(z_*) - g_t^i\|^2 \Big| \mathcal{F}_{t-1}\right) \leqslant (1-\rho)\sum_{i\in\mathcal{I}}\frac{1}{\pi_i}\|B_i(z_*) - g_{t-1}^i\|^2 + \rho\sum_{i\in\mathcal{I}}\frac{1}{\pi_i}\|B_i(z_*) - B_i(z_{t-1})\|^2$$

$$\leqslant (1-\rho)\sum_{i\in\mathcal{I}}\frac{1}{\pi_i}\|B_i(z_*) - g_{t-1}^i\|^2 + \rho\bar{L}^2\|z_{t-1} - z_*\|^2.$$

Therefore, overall, we have, for a scalar $b > 0$ to be chosen later:

$$\mathbb{E}\left(\|z_t - z_*\|^2 + b\sum_{i\in\mathcal{I}}\frac{1}{\pi_i}\|B_i(z_*) - g_t^i\|^2 \Big| \mathcal{F}_{t-1}\right)$$

$$\leqslant \left(1 - 2\eta + \eta^2 + \eta^2\frac{L^2}{\mu^2} + \frac{(1+a)\eta^2}{\mu^2}\frac{\bar{L}^2}{m} + b\rho\bar{L}^2\right)\|z_{t-1} - z_*\|^2$$

$$+ b\left(1 - \rho + b^{-1}\frac{(1+a^{-1})\eta^2}{m\mu^2}\right)\sum_{i\in\mathcal{I}}\frac{1}{\pi_i}\|B_i(z_*) - g_{t-1}^i\|^2.$$

If we take $a = 2$, $\eta = \frac{1}{\max\{\frac{3|\mathcal{I}|}{2m}, 1 + \frac{L^2}{\mu^2} + 3\frac{\bar{L}^2}{m\mu^2}\}}$, which corresponds to $\sigma = \frac{1}{\mu}\frac{\eta}{1-\eta} = \frac{\mu}{\max\{\frac{3|\mathcal{I}|}{2m} - 1, \frac{L^2}{\mu^2} + 3\frac{\bar{L}^2}{m\mu^2}\}}$, with $b\rho\bar{L}^2 = \frac{3\eta}{4}$, then we get the bound (using $\eta \leqslant 1/(\bar{L}^2/(3m))$):

$$(1 - \frac{\eta}{4})\|z_{t-1} - z_*\|^2 + (1 - \frac{\rho}{3})\sum_{i\in\mathcal{I}}\frac{1}{\pi_i}\|B_i(z_*) - g_{t-1}^i\|^2,$$

which shows that the function $(z, g) \mapsto \|z - z_*\|^2 + b\sum_{i\in\mathcal{I}}\frac{1}{\pi_i}\|B_i(z_*) - g^i\|^2$ is a good Lyapunov function for the problem that shrinks geometrically in expectation (it resembles the one from convex minimization, but without the need for function values).

Finally, since we assume that $m \leqslant |\mathcal{I}|$, we have $\rho = 1 - (1 - 1/|\mathcal{I}|)^m \geqslant 1 - \exp(-m/|\mathcal{I}|) \geqslant m/(2|\mathcal{I}|)$. This leads to, after $t$ iterations

$$\mathbb{E}\|z_t - z_*\|^2 \leqslant (1 - \min\{\frac{\eta}{4}, \frac{m}{6|\mathcal{I}|}\})^t\left[\|z_0 - z_*\|^2 + \frac{3\eta}{4\rho\bar{L}^2}\sum_{i\in\mathcal{I}}\frac{1}{\pi_i}\|B_i(z_*) - B_i(z_0)\|^2\right].$$

We have $\eta \leqslant 2m/(3|\mathfrak{I}|)$ and $3\eta/(4\rho) \leqslant \frac{3}{4}\frac{2m}{3|\mathfrak{I}|}\frac{2|\mathfrak{I}|}{m} \leqslant 1$, leading to

$$\mathbb{E}\|z_t - z_*\|^2 \leqslant 2(1 - \frac{\eta}{4})^t \|z_0 - z_*\|^2,$$

which is the desired result.

Note that we get the same overall running-time complexity than for SVRG.

**Factored splits.** Note that when applying to saddle-points with factored splits, we need to use a Lyapunov function that considers these splits. The only difference is to treat separately the two parts of the vectors, leading to replacing everywhere $|\mathfrak{I}|$ by $\max\{|\mathfrak{I}|, |\mathfrak{K}|\}$.

### D.3 Acceleration

We also consider in this section a proof based on monotone operators. We first give the algorithm for saddle-point problems.

**Algorithms for saddle-point problems.** At each iteration, we need solve the problem in Eq. (4) of the main paper, with the SVRG algorithm applied to $\tilde{K}(x,y) = K(x,y) - \lambda\tau x^\top \bar{x} + \gamma\tau y^\top \bar{y}$, and $\tilde{M}(x,y) = M(x,y) + \frac{\lambda\tau}{2}\|x\|^2 - \frac{\gamma\tau}{2}\|y\|^2$. These functions lead to constants $\tilde{\lambda} = \lambda(1+\tau)$, $\tilde{\gamma} = \gamma(1+\tau)$ and $\tilde{L} = L/(1+\tau)$, $\tilde{\sigma} = \sigma(1+\tau)^2$. We thus get the iteration, for a single selected operator,

$$(x,y) \leftarrow \text{prox}_{\tilde{M}}^{\tilde{\sigma}}\big[(x,y) - \tilde{\sigma}\big(\begin{smallmatrix} 1/\tilde{\lambda} & 0 \\ 0 & 1/\tilde{\gamma} \end{smallmatrix}\big)\big(\tilde{B}(\tilde{x},\tilde{y}) + \{\frac{1}{\pi_i}\tilde{B}_i(x,y) - \frac{1}{\pi_i}\tilde{B}_i(\tilde{x},\tilde{y})\}\big)\big].$$

A short calculation shows that $\text{prox}_{\tilde{M}}^{\tilde{\sigma}}(x,y) = \text{prox}_M^{\sigma(1+\tau)/(1+\sigma\tau(1+\tau))}((x,y)/(1+\sigma\tau(1+\tau)))$, leading to the update (with $\sigma$ the step-size from the regular SVRG algorithm in Section 3):

$$(x,y) \leftarrow \text{prox}_{\tilde{M}}^{\tilde{\sigma}}\big[(x,y) + \sigma\tau(1+\tau)(\bar{x},\bar{y}) - \sigma(1+\tau)\big(\begin{smallmatrix} 1/\lambda & 0 \\ 0 & 1/\gamma \end{smallmatrix}\big)\big(B(\tilde{x},\tilde{y}) + \{\frac{1}{\pi_i}\tilde{B}_i(x,y) - \frac{1}{\pi_i}\tilde{B}_i(\tilde{x},\tilde{y})\}\big)\big].$$

This leads to Algorithm 3, where differences with the SVRG algorithm, e.g., Algorithm 1, are highlighted in red. Given the value of $\tau$, the estimate $(\bar{x}, \bar{y})$ is updated every $\log(1+\tau)$ epochs of SVRG. While this leads to a provably better convergence rate, in practice, this causes the algorithm to waste time solving with too high precision the modified problem. We have used the simple heuristic of changing $(\bar{x}, \bar{y})$ one epoch after the primal-dual gap has been reduced from the previous change of $(\bar{x}, \bar{y})$.

---

**Algorithm 3** Accelerated Stochastic Variance Reduction for Saddle Points

---

**Input:** Functions $(K_i)_i$, $M$, probabilities $(\pi_i)_i$, smoothness $\bar{L}(\pi)$ and $L$, iterate $(x,y)$
　　　　number of epochs $v$, number of updates per iteration $m$, acceleration factor $\tau$

　Set $\sigma = \big[L^2 + 3\bar{L}^2/m\big]^{-1}$ and $(\bar{x}, \bar{y}) = (x,y)$
　**for** $u = 1$ to $v$ **do**
　　If $u = 0 \mod \lceil 2 + 2\log(1+\tau)/(\log 4/3)\rceil$, set $(\bar{x}, \bar{y}) = (\tilde{x}, \tilde{y})$
　　Initialize $(\tilde{x}, \tilde{y}) = (x,y)$ and compute $B(\tilde{x}, \tilde{y})$
　　**for** $k = 1$ to $\log 4 \times (L^2 + 3\bar{L}^2/m)(1+\tau)^2$ **do**
　　　Sample $i_1, \dots, i_m \in \mathfrak{I}$ from probability vector $(\pi_i)_i$ with replacement
　　　$z \leftarrow (x,y) + \sigma\tau(1+\tau)(\bar{x},\bar{y}) - \sigma(1+\tau)\big(\begin{smallmatrix} 1/\lambda & 0 \\ 0 & 1/\gamma \end{smallmatrix}\big)\big(B(\tilde{x},\tilde{y}) + \{\frac{1}{\pi_i}\tilde{B}_i(x,y) - \frac{1}{\pi_i}\tilde{B}_i(\tilde{x},\tilde{y})\}\big)$
　　　$(x,y) \leftarrow \text{prox}_M^{\sigma(1+\tau)/(1+\sigma\tau(1+\tau))}(z/(1+\sigma\tau(1+\tau)))$
　　**end for**
　**end for**
**Output:** Approximate solution $(x,y)$

---

**Proof of Theorem 3 using monotone operators.** We consider $\tau \geqslant 0$, and we consider the following algorithm, which is the transposition of the algorithm presented above. We consider a mini-batch $m = 1$ for simplicity. We consider a set of SVRG epochs, where $\bar{z}$ remains fixed. These epochs are initialized by $\tilde{z} = \bar{z}$.

For each SVRG epoch, given $\bar{z}$ and $\tilde{z}$, and starting from $z = \tilde{z}$, we run $t$ iterations of:

$$z \leftarrow (I + \sigma(\tau I + A))^{-1}\big(z - \sigma[B\tilde{z} + \frac{1}{\pi_i}(B_i z - B_i \tilde{z}) - \tau\bar{z}]\big),$$

and then update $\tilde{z}$ as $z$ at the end of the SVRG epoch. It corresponds exactly to running the SVRG algorithm to find $(\tau I + A + B)^{-1}(\tau\bar{z})$ approximately, we know from the proof of Theorem 1 that after $\log 4\big(1 + \frac{L^2}{\mu^2(1+\tau)^2} + \frac{L^2}{\mu^2(1+\tau)^2}\big)$ iterations, we have an iterate $z$ such that $\mathbb{E}\|z - (\tau I + A + B)^{-1}(\tau\bar{z})\|^2 \leqslant \frac{3}{4}\mathbb{E}\|\tilde{z} - (\tau I + A + B)^{-1}(\tau\bar{z})\|^2$. Thus, if we run $s$ epochs where we update $\tilde{z}$ (but not $\bar{z}$) at each start of epoch, we get an iterate $z$ such that $\mathbb{E}\|z - (\tau I + A + B)^{-1}(\tau\bar{z})\|^2 \leqslant (\frac{3}{4})^s \mathbb{E}\|\bar{z} - (\tau I + A + B)^{-1}(\tau\bar{z})\|^2$, and thus

$$
\begin{aligned}
&\mathbb{E}\|z - (\tau I + A + B)^{-1}(\tau\bar{z})\|^2 \\
\leqslant \quad & \left(\frac{3}{4}\right)^s \mathbb{E}\|\bar{z} - (\tau I + A + B)^{-1}(\tau\bar{z})\|^2 \\
= \quad & \left(\frac{3}{4}\right)^s \mathbb{E}\|\bar{z} - z_* - (\tau I + A + B)^{-1}(\tau\bar{z}) + (\tau I + A + B)^{-1}(\tau z_*)\|^2 \\
& \qquad\qquad \text{using } z_* = (\tau I + A + B)^{-1}(\tau z_*), \\
= \quad & \left(\frac{3}{4}\right)^s \mathbb{E}\|\bar{z} - z_* - (I + \tau^{-1}(A+B))^{-1}(\bar{z}) + (I + \tau^{-1}(A+B))^{-1}(z_*)\|^2.
\end{aligned}
$$

We may now use the fact that for any multi-valued maximal monotone operator $C$, $I - (I+C)^{-1} = (I + C^{-1})^{-1}$, which shows that $I - (I+C)^{-1}$ is 1-Lipschitz-continuous. Thus, after $s$ epochs of SVRG,

$$\mathbb{E}\|z - (\tau I + A + B)^{-1}(\tau\bar{z})\|^2 \quad \leqslant \quad \left(\frac{3}{4}\right)^s \mathbb{E}\|\bar{z} - z_*\|^2.$$

This implies, by Minkowski's inequality,

$$
\begin{aligned}
& (\mathbb{E}\|z - z_*\|^2)^{1/2} \\
\leqslant \quad & (\mathbb{E}\|z - (\tau I + A + B)^{-1}(\tau\bar{z})\|^2)^{1/2} + (\mathbb{E}\|(\tau I + A + B)^{-1}(\tau\bar{z}) - z_*\|^2)^{1/2} \\
\leqslant \quad & \left(\frac{3}{4}\right)^{s/2} (\mathbb{E}\|\bar{z} - z_*\|^2)^{1/2} + (\mathbb{E}\|(\tau I + A + B)^{-1}(\tau\bar{z}) - (\tau I + A + B)^{-1}(\tau z_*)\|^2)^{1/2} \\
= \quad & \left(\frac{3}{4}\right)^{s/2} (\mathbb{E}\|\bar{z} - z_*\|^2)^{1/2} + (\mathbb{E}\|(I + \tau^{-1}(A+B))^{-1}(\bar{z}) - (I + \tau^{-1}(A+B))^{-1}(z_*)\|^2)^{1/2} \\
\leqslant \quad & \left(\frac{3}{4}\right)^{s/2} (\mathbb{E}\|\bar{z} - z_*\|^2)^{1/2} + \frac{1}{1 + \tau^{-1}\mu}(\mathbb{E}\|\bar{z} - z_*\|^2)^{1/2} \\
= \quad & \left(\frac{3}{4}\right)^{s/2} (\mathbb{E}\|\bar{z} - z_*\|^2)^{1/2} + \frac{\tau}{\tau + \mu}(\mathbb{E}\|\bar{z} - z_*\|^2)^{1/2},
\end{aligned}
$$

using the fact that the contractivity of resolvents of strongly monotone operators. Thus after $s = 2 + 2\frac{\log(1+\frac{\tau}{\mu})}{\log\frac{4}{3}}$, we get a decrease by $(1 - \frac{\mu}{\tau+\mu})$, and thus the desired result.

## D.4 Factored splits and bi-linear models

In the table below, we report the running-time complexity for the factored splits which we used in simulations. Note that SAGA and SVRG then have different bounds. Moreover, all these schemes are adapted when $n$ is close to $d$. For $n$ much different from $d$, one could imagine to (a) either complete with zeros or (b) to regroup the data in the larger dimension so that we get as many blocks as for the lower dimension.

| Algorithms | Complexity |
|---|---|
| Stochastic FB-non-uniform | $(1/\varepsilon) \times \left( \ \max\{n,d\}\|K\|_F^2/(\lambda\gamma) \hspace{2.5cm} \right)$ |
| Stochastic FB-uniform | $(1/\varepsilon) \times \left( \ \max\{n,d\}^2\|K\|_{\max}^2/(\lambda\gamma) \hspace{1.8cm} \right)$ |
| SVRG-uniform | $\log(1/\varepsilon) \times \left( \ nd + \max\{n,d\}^2\|K\|_{\max}^2/(\lambda\gamma) \hspace{0.8cm} \right)$ |
| SAGA-uniform | $\log(1/\varepsilon) \times \left( \ \max\{n,d\}^2 + \max\{n,d\}^2\|K\|_{\max}^2/(\lambda\gamma) \ \right)$ |
| SVRG-non-uniform | $\log(1/\varepsilon) \times \left( \ nd + \max\{n,d\}\|K\|_F^2/(\lambda\gamma) \hspace{1.0cm} \right)$ |
| SAGA-non-uniform | $\log(1/\varepsilon) \times \left( \ \max\{n,d\}^2 + \max\{n,d\}\|K\|_F^2/(\lambda\gamma) \ \right)$ |
| SVRG-non-uniform-acc. | $\log(1/\varepsilon) \times \left( \ nd + \max\{n,d\}^{3/2}\|K\|_F/\sqrt{\lambda\gamma} \hspace{0.6cm} \right)$ |

Table 2: Summary of convergence results for the strongly $(\lambda,\gamma)$-convex-concave bilinear saddle-point problem with matrix $K$ and factored splits, with access to a single row and a single column per iteration. The difference with the individual splits from Table 1 is highlighted in red.

# E  Surrogate to Area Under the ROC Curve

We consider the following loss function on $\mathbb{R}^n$, given a vector of positive and negative labels, which corresponds to a convex surrogate to the number of misclassified pairs [13, 14]:

$$
\begin{aligned}
\ell(u) &= \frac{1}{2n_+n_-} \sum_{i_+\in I_+} \sum_{i_-\in I_-} (1 - u_{i_-} + u_{i_+})^2 \\
&= \frac{1}{2n_+n_-} \sum_{i_+\in I_+} \sum_{i_-\in I_-} \left\{ 1 + u_{i_-}^2 + u_{i_+}^2 - 2u_{i_-} + 2u_{i_+} - 2u_{i_-}u_{i_+} \right\} \\
&= \frac{1}{2} + \frac{1}{n_+}\sum_{i_+\in I_+} u_{i_+} - \frac{1}{n_-}\sum_{i_-\in I_-} u_{i_-} + \frac{1}{2n_-}\sum_{i_-\in I_-} u_{i_-}^2 + \frac{1}{2n_+}\sum_{i_+\in I_+} u_{i_+}^2 - \frac{1}{n_+n_-}\sum_{i_+\in I_+}\sum_{i_-\in I_-} u_{i_-}u_{i_+} \\
&= \frac{1}{2} + \frac{1}{n_+}e_+^\top u - \frac{1}{n_-}e_-^\top u + \frac{1}{2}u^\top \mathrm{Diag}(\frac{1}{n_+}e_+ + \frac{1}{n_-}e_-)u - \frac{1}{2n_+n_-}u^\top(e_+e_-^\top + e_-e_+^\top)u \\
&= \frac{1}{2} - a^\top u + \frac{1}{2}u^\top A u,
\end{aligned}
$$

with $e_+ \in \mathbb{R}^n$ the indicator vector of $I_+$ and $e_- \in \mathbb{R}^n$ the indicator vector of $I_-$. We have $A = \mathrm{Diag}(\frac{1}{n_+}e_+ + \frac{1}{n_-}e_-) - \frac{1}{n_+n_-}\left[e_+e_-^\top + e_-e_+^\top\right]$ and $a = e_+/n_+ - e_-/n_-$. A short calculation shows that the largest eigenvalue of $A$ is $\frac{1}{M} = \frac{1}{n_+} + \frac{1}{n_-}$.

We consider the function $h(u) = \frac{1}{2}u^\top A u$. It is $(1/M)$-smooth, its Fenchel conjugate is equal to

$$\frac{1}{2}v^\top A^{-1}v,$$

and our function $g$ will be equal to $v \mapsto \frac{1}{2}v^\top A^{-1}v - \frac{M}{2}\|v\|^2$. Given that $1$ is a singular vector of $A$, $g(v)$ is finite only when $v^\top 1_n = 0$.

We need to be able to compute $g(v)$, i.e., solve the system $A^{-1}v$, and to compute the the proximal operator

$$\min_v \frac{1}{2}\|v - v_0\|^2 + \sigma g(v) = \min_v \frac{1}{2}\|v - v_0\|^2 + \frac{\sigma}{2}v^\top(A^{-1} - MI)v,$$

which leads to to the system: $(A^{-1} - MI + \sigma^{-1}I)v = \sigma^{-1}v_0$, which is equivalent to: $(I - MA + \sigma^{-1}A)v = \sigma^{-1}Av_0$ We thus need to compute efficiently $Aw$, and $(I + \kappa A)^{-1}w$ with $\kappa > -M$. We

have

$$I + \kappa A = \text{Diag}((1 + \kappa/n_+)e_+ + (1 + \kappa/n_-)e_-) - \frac{\kappa}{n_+ n_-}\left[e_+ e_-^\top + e_- e_+^\top\right]$$

$$= \text{Diag}((1 + \kappa/n_+)e_+ + (1 + \kappa/n_-)e_-)^{1/2}$$
$$\left[I - \frac{\kappa}{n_+ n_-}\left(\left[\frac{1}{\sqrt{1 + \kappa/n_+}}e_+\right]\left[\frac{1}{\sqrt{1 + \kappa/n_-}}e_-\right]^\top - \left[\frac{1}{\sqrt{1 + \kappa/n_-}}e_-\right]\left[\frac{1}{\sqrt{1 + \kappa/n_+}}e_+\right]^\top\right)\right]$$
$$\text{Diag}((1 + \kappa/n_+)e_+ + (1 + \kappa/n_-)e_-)^{1/2}$$
$$= D^{1/2}(I - \alpha u_+ u_-^\top - \alpha u_- u_+^\top)D^{1/2},$$

with $u_+^\top u_- = 0$ and $u_+ = \frac{e_+}{\sqrt{n_+}}, u_- = \frac{e_-}{\sqrt{n_-}}$ of norm 1 and $D = \text{Diag}((1 + \kappa/n_+)e_+ + (1 + \kappa/n_-)e_-)$. We have:

$$I - \alpha u_+ u_-^\top - \alpha u_- u_+^\top = I - u_+ u_+^\top - u_- u_-^\top + (u_+, u_-)\begin{pmatrix} 1 & -\alpha \\ -\alpha & 1 \end{pmatrix}(u_+, u_-)^\top$$

$$(I - \alpha u_+ u_-^\top - \alpha u_- u_+^\top)^{-1} = I - u_+ u_+^\top - u_- u_-^\top + \frac{1}{1 - \alpha^2}(u_+, u_-)\begin{pmatrix} 1 & \alpha \\ \alpha & 1 \end{pmatrix}(u_+, u_-)^\top$$

$$= I + (1/(1 - \alpha^2) - 1)u_+ u_+^\top + (1/(1 - \alpha^2) - 1)u_- u_-^\top + \frac{\alpha}{1 - \alpha^2}(u_+ u_-^\top + u_- u_+^\top)$$

$$= I + (1/(1 - \alpha^2) - 1)\frac{1}{n_+}e_+ e_+^\top + (1/(1 - \alpha^2) - 1)\frac{1}{n_-}e_- e_-^\top$$

$$+ \frac{\alpha}{1 - \alpha^2}\frac{1}{\sqrt{n_+ n_-}}(e_+ e_-^\top + e_- e_+^\top).$$

We have here $\alpha = \frac{\kappa}{n_+ n_-}\sqrt{\frac{n_+}{1 + \kappa/n_+}}\sqrt{\frac{n_-}{1 + \kappa/n_-}}$. Thus

$$(I + \kappa A)^{-1} = D^{-1/2}\left[I - u_+ u_+^\top - u_- u_-^\top + \frac{1}{1 - \alpha^2}(u_+, u_-)\begin{pmatrix} 1 & \alpha \\ \alpha & 1 \end{pmatrix}(u_+, u_-)^\top\right]D^{-1/2},$$

which can be done in $O(n)$.

Moreover, we have

$$A = \text{Diag}((1/n_+)e_+ + (1/n_-)e_-)^{1/2}$$
$$\left[I - \frac{1}{n_+ n_-}\left(\left[\sqrt{n_+}e_+\right]\left[\sqrt{n_-}e_-\right]^\top - \left[\sqrt{n_-}e_-\right]\left[\sqrt{n_+}e_+\right]^\top\right)\right]$$
$$\text{Diag}((1/n_+)e_+ + (1/n_-)e_-)^{1/2}$$
$$= D^{1/2}(I - u_+ u_-^\top - u_- u_+^\top)D^{1/2},$$

with $u_+^\top u_- = 0$ and $u_+, u_-$ of norm 1. Thus we have

$$I - u_+ u_-^\top - u_- u_+^\top = I - u_+ u_+^\top - u_- u_-^\top + (u_+, u_-)\begin{pmatrix} 1 & -1 \\ -1 & 1 \end{pmatrix}(u_+, u_-)^\top$$

$$(I - u_+ u_-^\top - u_- u_+^\top)^{-1} = I - u_+ u_+^\top - u_- u_-^\top + \frac{1}{0}(u_+, u_-)\begin{pmatrix} 1 & 1 \\ 1 & 1 \end{pmatrix}(u_+, u_-)^\top.$$

Thus, if $v^\top 1_n = 0$, we get:

$$v^\top A^{-1} v = v^\top \text{Diag}(n_+ e_+ + n_- e_-)v - (v^\top e_+)^2 - (v^\top e_-)^2,$$

which has running-time complexity $O(n)$.

**Optimization problem.** With a regularizer $f(x) + \frac{\lambda}{2}\|x\|^2$, we obtain the problem:

$$\min_{x \in \mathbb{R}^d} \frac{\lambda}{2}\|x\|^2 + f(x) + \frac{1}{2} - a^\top Kx + \frac{1}{2}(Kx)^\top A(Kx)$$

$$\min_{x \in \mathbb{R}^d} \max_{y \in \mathbb{R}^n} \frac{\lambda}{2}\|x\|^2 + f(x) + \frac{1}{2} - a^\top Kx + y^\top Kx - \frac{M}{2}\|y\|^2 - \frac{1}{2}y^\top (A^{-1} - MI)y,$$

with $g(y) = \frac{1}{2}y^\top (A^{-1} - MI)y$.

# F    Additional Experimental Results

We complement the results of the main paper in several ways: (a) by providing all test losses, the distance to optimum $\Omega(x - x_*, y - y_*)$ in log-scale, as well as the primal-dual gaps in log-scale, as a function of the number of passes on the data. We consider the three machine learning settings:

– Figure 1: `sido` dataset, AUC loss and cluster norm (plus squared-norm) regularizer (both non separable).
– Figure 2: `sido` dataset, square loss and $\ell_1$-norm (plus squared-norm) regularizer (both separable).
– Figure 3: `rcv1` dataset, square loss and $\ell_1$-norm (plus squared-norm) regularizer (both separable).

We consider the following methods in all cases (all methods are run with the step-sizes proposed in their respective convergence analysis):

– fb-acc: accelerated forward-backward saddle-point method from Section 2.2,
– fb-sto: stochastic forward-backward saddle-point method from Section 2.3,
– saga: our new algorithm from Section 4, with non-uniform sampling, and sampling of a single row and column per iteration,
– saga (unif): our new algorithm from Section 4, with uniform sampling, and sampling of a single row and column per iteration,
– svrg: our new algorithm from Section 3, with non-uniform sampling, and sampling of a single row and column per iteration,
– svrg-acc: our new accelerated algorithm from Section 3, with non-uniform sampling, and sampling of a single row and column per iteration,
– fba-primal: accelerated proximal method [10], which can be applied to the primal version of our problem (which is the sum of a smooth term and a strongly convex term).

Moreover, for the separable cases, we add:

– saga-primal: SAGA with non-uniform sampling [25], which can only be run with separable losses.

We can make the following observations:

– Non-uniform sampling is key to good performance.
– The distance to optimum (left plots) exhibits a clear linear convergence behavior (which is predicted by our analysis), which is not the case for the primal-dual gap, which does converge, but more erratically. It would be interesting to provide bounds for these as well.
– When $\lambda$ decreases (bottom plots, more ill-conditioned problems), the gains of accelerated methods with respect to non-accelerated ones are unsurprisingly larger. Note that for two out of three settings, the final test loss is smaller for the smaller regularization, and non-accelerated methods need more passes on the data to reach good testing losses.
– Primal methods which are not using separability (here "fba-primal") can be run on all instances, but are not competitive. Note that in some situations, they achieve early on good performances (e.g., Figure 2), before getting caught up by stochastic-variance-reduced saddle-point techniques (note also that since these are not primal-dual methods, we compute dual candidates through the gradient of the smooth loss functions, which is potentially disadvantageous).
– Primal methods that use separability (here "saga-primal") cannot be run on non-separable problems, but when they can run, they are still significantly faster than our saddle-point techniques. We believe that this is partly due to adaptivity to strong convexity (the convergence bounds for the two sets of techniques are the same for this problem).

Figure 1: `sido` dataset. Top: $\lambda = \lambda_0 = \|K\|_F^2/n^2$, Bottom: $\lambda = \lambda_0/10 = \frac{1}{10}\|K\|_F^2/n^2$. AUC loss and cluster-norm regularizer. Distances to optimum, primal-dual gaps and test losses, as a function of the number of passes on the data. Note that the primal SAGA (with non-uniform sampling) cannot be used because the loss is not separable. Best seen in color.

Figure 2: `sido` dataset. Top: $\lambda = \lambda_0 = \|K\|_F^2/n^2$, Bottom: $\lambda = \lambda_0/10 = \frac{1}{10}\|K\|_F^2/n^2$. Squared loss, with $\ell_1$-regularizer. Distances to optimum, primal-dual gaps and test losses, as a function of the number of passes on the data. Note that the primal SAGA (with non-uniform sampling) can only be used because the loss is separable. Best seen in color.

Figure 3: rcv1 dataset. Top: $\lambda = \lambda_0 = \|K\|_F^2/n^2$, Bottom: $\lambda = \lambda_0/10 = \frac{1}{10}\|K\|_F^2/n^2$. Squared loss, with $\ell_1$-regularizer. Distances to optimum, primal-dual gaps and test losses, as a function of the number of passes on the data. Note that the primal SAGA (with non-uniform sampling) can only be used because the loss is separable. Best seen in color.