[Reviews · NeurIPS 2016]

Reviewer 1

Summary

This paper considers saddle-point problems with a convex-concave property. The authors show how this problem can be addressed using existing stochastic variance reduced techniques (such as SVRG and SAGA) coupled with a proximal operator. They study these algorithms in the monotone operator setting showing linear convergence. They also propose an accelerated variant as well as analyze a non-uniform sampling scheme.

Qualitative Assessment

Novelty/originality: The contributions of this paper are significant. The theoretical analysis is thorough and reveals some new insights for saddle-point problems. The authors propose many extensions in this paper, the most significant one being that their method applies to non-separable functions. Technical quality: Although I really like the theoretical contributions of this paper, I was nevertheless quite disappointed in the way it is presented. I think the authors could have made the paper more accessible by making the connection to monotone operators more clear. This appears as an extension in Section 6 but the analysis provided by the authors is based on monotone operators. On the experimental side, I would have liked to see experimental results on more than two datasets. In the case of separable functions, the authors should also have compared to accelerated methods such as this stochastic variant of Chambolle-Pock algorithm: http://www.jmlr.org/proceedings/papers/v37/zhanga15.pdf Please add this reference to your submission. There are a few points that I would like the authors should clarify: 1. How strong are the assumptions A-C, particularly assumption (A). For example, I wonder whether their assumptions hold for solving the saddle-point problem induced by SVM. 2. I can not fully understand the differences between stochastic primal-dual method and their suggested method. Can the authors elaborate on this point? 3. In theorem 2, I am not sure about constant \mu. Is it the monotonicity constant mentioned in appendix? How can we compare this result with the result of theorem 1? 4. The transition from assumption (A)-(C) to strong monotonicity (in appendix) is confusing. Do assumptions (A)-(C) imply monotonicity? Minor issues: Some mix-up with the constant L used as the condition number and the Lipschitz constant: “The quantity L represents the condition number of the problem” and “we need the Lipschitz constant L” Forward-Backward algorithm: please add a reference

Confidence in this Review

3-Expert (read the paper in detail, know the area, quite certain of my opinion)


Reviewer 2

Summary

This work considers the problem of minimizing convex-concave saddle problem and extend the recent variance reduction methods for solving it. The convergence analysis of the proposed methods are provided. The authors shows that the methods can be easily accelerated by using the extension of the catalyst framework. Extension to the monotone operator and numerical result are also provided.

Qualitative Assessment

The strong convex assumptions on both primal and dual problem is every strong. It should be only primal problem or dual problem is strongly convex. More numerical example should be added.

Confidence in this Review

2-Confident (read it all; understood it all reasonably well)


Reviewer 3

Summary

The paper extends several of the recent incremental algorithms (SVRG, SAGA, and their accelerated variants via Catalyst) to solve strongly convex-concave saddle point problems with finite sum structure, which could result from two types of splits, individual or factored splits. The authors provide a thorough study on the complexity of these algorithms under different splits and different (uniform/nonuniform) sampling strategies. Experiment on a special binary classification example show that nonuniform sampling and acceleration are critical to beat the competitive batch algorithms.

Qualitative Assessment

While the idea of extending variance reduction and incremental algorithms to saddle point problem might sound straightforward, the results established here is notably novel and the technicality is surely non-trivial. For the important bilinear saddle point problem, the authors show clear comparison among batch, stochastic, and incremental FB algorithms, which share some resemblance to what's usually know for the convex minimization case, and it all makes sense to me. However, here is my major concern. While solving the saddle point problem cures the difficulty of non-separability of loss functions, the proposed incremental algorithms seemingly suffer from a fundamental bottleneck, i.e. O(n+d) cost for computing the proximal operator at each iteration. As a result, the overall complexity, even for the accelerated SVRG with non-uniform sampling, as shown in Table 1 and 2, is not necessarily better than the accelerated batch version. Especially, when the number of data n is substantially larger that its dimension d, which is the case for most machine learning applications, the complexity under factored splits (later used in simulation) is actually worse than the batch FB algorithms, as shown in Table 2. Unlike what's usually known for the convex minimization, I did not see a clear substantial advantage of using variance reduction here, from a theoretical point of view. I would suggest the authors to provide an in-depth discussion on this matter and clarify the following questions i) is factored splits (theoretically) worse than individual splits, ii) when is the accelerated SVRG favorable against the batch FB, iii) is the dependence on n, d, tight enough? == post-rebuttal answer== I have read the author's rebuttal and is satisfied with the response. My scores remain the same.

Confidence in this Review

2-Confident (read it all; understood it all reasonably well)


Reviewer 4

Summary

This paper extends the stochastic variance reduction techniques SVRG and SAGA to deal with strongly convex-concave saddle-point problems where linear convergence is proved. Both theoretical and empirical results demonstrate the effectiveness of the variance reduction method on the saddle-point problems.

Qualitative Assessment

This paper extends the stochastic variance reduction techniques SVRG and SAGA to deal with strongly convex-concave saddle-point problems. The proposed algorithms can apply to a number of non-separable saddle-point problems, where both the loss (e.g., structured output prediction) and regularizer (e.g., fusion term) can be non-separable. An important contribution is that, different from the previous convergence analysis on convex problems, the analysis on saddle-point problem is much more complex and the authors adopt the monotone operators to prove the convergence. The analysis also shows that the extended SVRG and SAGA algorithms apply to a wider class of problems such as the variational inequality in game theory. Considering a stochastic approach, the authors propose two different splits of the gradient operator, i.e., the element-wise split and factored split, for the SVRG and SAGA respectively. The variance reduction works on the stochastic forward-backward algorithms which are commonly used for saddle-point problem. Non-uniform sampling is considered as well and both theoretical and empirical results show that variance reduction with non-uniform sampling is superior to that with uniform sampling. Experimental results in both the main article and the appendix provide sufficient evidence of the theoretical results and demonstrate the effectiveness of the proposed methods on both non-separable loss and regularizer. Some minor comments: As acceleration in Section 5, the authors follow [8] to incorporate an additional regularization term to push the update similar to an iterate point. It is not clear that how to adaptively update this iterate point, namely (\bar{x}, \bar{y}), to achieve speed-up. A convergence and acceleration proof is provided for SVRG. Will the same theoretical results exist for SAGA? The smooth function K(.) and the design matrix K should use different symbols to avoid confusion.

Confidence in this Review

1-Less confident (might not have understood significant parts)


Reviewer 5

Summary

The authors use variance reduction algorithms like SVRG and SAGA to propose the first linearly convergent algorithms for saddle point problems. Since the convex minimization analysis does not apply for the saddle point problems, the main contribution of the paper is using monotone operators to prove convergence. A consequence of this is that the analysis applies to a wider class of problems such as variational inequality problems. This method can be used to tackle non-separable loss functions and regularizers by considering the saddle point problem.

Qualitative Assessment

The paper proposes the first algorithms with a linear convergence rate for saddle point problems. This is an important problem to solve since it helps to tackle non-separable loss functions and regularizers, which occur in a number of machine learning models. The proof technique of interpreting saddle point problems as finding zeros of a monotone operator is interesting, and potentially widely applicable. Simpler proofs for algorithms like the FB splitting (as well as its stochastic version) are also provided. The proposed algorithms with non-uniform sampling are competitive, and in most cases, better than previous alternatives. The paper overall extends the field in an important direction, and the analysis approach is interesting. Since the main contribution of the paper is in the proof technique (the algorithms itself are quite simple extensions of the original algorithms), the authors could consider putting in more details about the proof technique, for example by including a proof sketch, in the main paper, if space permits. The paper is mostly clearly presented and well-written. However, I found the paper a bit unclear on certain issues. For example: 1. What kind of non-uniform sampling is used for problems other than bilinear saddle-point problems, i.e., what is \pi? 2. How is the step size \sigma set in the experiments? Do the authors have any intuition on why non-uniform sampling seems so essential for saddle point problems, and not necessarily for convex minimization problems? The paper might benefit from a discussion on this issue. Minor comments: 1. Typo on line 78: k -> d 2. Typo on line 121 in the second equality (extra y_{t-1} term)

Confidence in this Review

2-Confident (read it all; understood it all reasonably well)